# SPKGDIAG: LEARNING SYMPTOM-LINKED PATIENT KNOWLEDGE GRAPHS VIA MULTI-HOP SIMILARITY MESSAGE PASSING FOR AUTOMATIC DIAGNOSIS

## ABSTRACT

Automated diagnostics in medicine leverage advanced algorithms to detect, analyze, and interpret medical conditions from data without human intervention. Existing systems predominantly focus on disease prediction, frequently neglecting the critical role of comprehensive symptom analysis. While some prior studies explored the reasoning capabilities of large language models (LLMs), they faced challenges in effectively integrating structured medical knowledge, limiting their ability to generate coherent and clinically relevant patient-centric representations. In this study, we propose SPKGDIAG, a novel framework that combines symptom extraction with patient-centric knowledge graph construction to enhance the accuracy and efficiency of disease diagnosis. We leverage LLM to automatically extract both implicit and explicit symptoms from patient-doctor conversations and construct a patient-centric knowledge graph with semantic embeddings. A multi-hop neighborhood sampling approach is used to capture common clinical symptoms by modeling both local patient-specific patterns and global population-level insights. Furthermore, we propose to use a specialized Message Passing Neural Network (MPNN) to process this graph structure for diagnosis prediction, aiming to balance semantic richness with structural relevance through message aggregation and self-projection mechanisms. We conducted extensive experiments on four benchmark datasets (MZ-4, MZ-10, Dxy, and Synthetic), achieving improvements of 1.4%, 4.4%, 2.0%, and 7.4% over the best existing methods, including RL, transform-based, and multi-department systems, respectively. Our model exhibited robust performance compared to recent baselines on a large-scale in-house dataset. The proposed framework provides an interpretable solution that enhances symptom-driven automatic diagnosis by integrating efficient natural language processing with structured medical reasoning.

## 1 INTRODUCTION

Automated diagnostics (AD) have gained significant research interest for their streamlined processes, ensuring safe implementation in sensitive healthcare settings while maintaining high diagnostic accuracy (Kao et al., 2018; Wei et al., 2018; Teixeira et al., 2021). These systems typically facilitate interaction between a diagnostic agent and a patient, with the agent collecting symptoms essential for diagnosis. The agent pursues two interdependent objectives: selecting the most informative symptoms to distinguish diseases and accurately identifying the disease. Generally framed as a multi-step inference process (Chen et al., 2022), this approach infers implicit symptoms from explicit ones before delivering a final diagnosis, closely reflecting real-world clinical workflows.

Figure 1 illustrates an automated diagnostic workflow that integrates the collection of explicit and implicit symptoms. The process begins with a patient's self-reported explicit symptoms (e.g., "cough" and "runny nose" for Ella). Through conversational natural dialogues, the diagnostic agent elicits additional implicit symptoms (e.g., "fever" and "sore throat") through conversation-based natural dialogues, simulating the discovery process described in previous studies (Wei et al., 2018; Teixeira et al., 2021; Chen et al., 2022; Hou et al., 2023). The set of collected symptoms is then used to determine the most likely disease (e.g., "flu" for Ella).

Figure 1: An illustration of the data utilized in the automated diagnostic process.

However, the development of automated diagnosis poses several challenges. *The first challenge is to combine LLM capabilities with structured medical knowledge.* Current approaches either rely on traditional machine learning (ML) approaches with limited inference capabilities (Wei et al., 2018; Xu et al., 2019) or use reinforcement learning (RL) frameworks that lack interpretability (Peng et al., 2018; Xia et al., 2020). Furthermore, Transformer-based methods such as DxFormer (Chen et al., 2023) and Diaformer (Chen et al., 2022) show improved performance but do not effectively leverage structured medical knowledge. While LLMs excel in natural language processing, they struggle with the systematic medical reasoning required for accurate diagnosis. Consequently, there is an urgent need for a model that effectively integrates the capabilities of LLM with structured medical knowledge, while preserving diagnostic accuracy by leveraging both explicit symptoms from patient self-reports and implicit symptoms derived through conversational interactions. *Another challenge is to construct a patient-centered knowledge representation.* Prior works mainly focus on symptom-disease mapping without considering the comprehensive patient picture. For example, KR-DS (Xu et al., 2019) and BSODA (He et al., 2022) incorporate a knowledge graph (KG) model that treats patients as isolated entities, neglecting the exploration of patient similarities and co-occurrence of symptoms, which are critical factors for achieving accurate diagnoses.

To address these challenges, we propose SPKGDIAG, a novel framework that combines symptom extraction with patient-centric knowledge graph construction for automatic diagnosis. We leverage LLMs to automatically extract both explicit and implicit symptoms from patient-doctor conversations and construct a patient-centric knowledge graph with semantic embeddings, capturing common clinical symptoms by modeling both local patient-specific patterns and global population-level insights through multi-hop neighborhood sampling. We introduce a specialized Message Passing Neural Network (MPNN) (Gilmer et al., 2017) to process this graph structure for diagnosis prediction, aiming to balance semantic richness with structural relevance through message aggregation and self-projection mechanisms. Our main contributions are summarized as follows.

- We introduced an integrated framework for automated diagnosis that synergistically combines LLMs for explicit and implicit symptom extraction with structured medical knowledge, addressing the limitations of existing methods, which often lack interpretability and fail to effectively leverage structured domain knowledge.

- We constructed a patient-centric knowledge graph that embeds symptom semantics and captures relationships between patients through symptom co-occurrence. Additionally, we applied an MPNN layer with multi-hop neighborhood sampling to model both individual patient characteristics and population-level patterns effectively.

- We conducted extensive experiments on four benchmark datasets (MZ-4, MZ-10, Dxy, and Synthetic) and an in-house dataset, demonstrating that SPKGDIAG outperformed state-of-the-art methods, with improved accuracy of up to 7.4%. These results highlighted the potential of the model in automated diagnosis based on interpretable symptoms.

## 2 RELATED WORK

Existing automated diagnostic techniques fall into four main categories: (1) conventional ML models, (2) RL-based approaches, (3) non-RL-based methods, and (4) knowledge-enhanced and graph-based approaches.

**Traditional approaches** like SVMs (Chang & Lin, 2011) incorporated explicit and implicit symptom features to establish diagnostic baselines but lacked the sequential decision-making capabilities essential for interactive diagnostics.

**RL-based techniques** have become increasingly prevalent in modeling diagnostic interactions. For instance, Wei et al. (2018) used deep Q-learning to detect implicit symptoms during consultations, while Peng et al. (2018) improved policy learning through reward shaping and symptom vector reconstruction. However, their reliance on simulated data limited real-world applicability. Hierarchical and knowledge-enhanced methods such as Zhong et al. (2022) and KR-DS (Xu et al., 2019) introduced multi-level decision structures and relation-aware symptom checking. Generative approaches like GAMP (Xia et al., 2020) further refined reward functions using adversarial learning. Yu et al. (2021) conducted a thorough study of the development and implementation of reinforcement learning in automated medical diagnosis. More recently, EIRAD (Yan et al., 2024) has advanced the field by incorporating medical knowledge graphs to guide reasoning, prune irrelevant nodes, and design reward signals that consider evidence sufficiency and diagnostic accuracy. However, RL methods still face challenges in data efficiency – a critical limitation in the data-scarce medical domain.

**Non-RL approaches** have emerged to address the stability and scalability challenges of RL-based models. BSODA (He et al., 2022) used knowledge-guided self-attention with information-theoretic objectives, while PPO-based models (Teixeira et al., 2021) leveraged GPT-2 for effective conversational modeling. Transformer-based designs have recently obtained cutting-edge outcomes. Dx-Former (Chen et al., 2023) utilized an encoder-decoder structure to separate symptom comprehension and disease prediction, whereas Diaformer (Chen et al., 2022) generated sequences for Alzheimer's disease (AD). CoAD (Wang et al., 2023) proposed a collaborative symptom-pathology generating technique using label expansion and sequence alignment. MTDiag (Hou et al., 2023) substituted unstable RL training with a multi-task classification framework enhanced with contrastive learning. These methods demonstrated strong predictive power but often overlooked structured medical knowledge, limiting clinical interpretability.

**Knowledge-enhanced and graph-based methods** have recently gained momentum by incorporating structured medical knowledge into diagnostic models. Zhang et al. (2023) combined Markov Logic Networks with LLM-extracted knowledge for interpretable, accurate diagnosis. KDPoG (Li & Ruan, 2024) leveraged heterogeneous GCNs and patient-oriented graphs to enhance symptom recall and diagnostic precision. Similarly, Tian et al. (2024) proposed a scalable, anti-forgetting framework that incrementally updated neural parameters in a weighted knowledge graph, enabling multi-departmental diagnosis. These approaches underscore the growing trend of leveraging structured knowledge to address the limitations of static or task-specific models in automatic diagnosis.

Unlike existing approaches such as GraphCare (Jiang et al., 2023), which build personalized graphs from structured EHRs, or multimodal contrastive learning frameworks (Lu et al., 2024) requiring rich multi-source data, our method is designed for a setting where only unstructured dialogues are available. Furthermore, compared to knowledge-seed or retrieval-based LLM prompting strategies (Wu et al., 2024), our proposed SPKGDIAG constructs a patient-centric graph grounded in real patient symptom patterns, using symptom-overlap edges and semantic similarity, followed by efficient two-hop neighborhood sampling. This design prioritizes interpretability, minimal reliance on external data, and robustness in low-resource environments.

## 3 METHODOLOGY

### 3.1 PRELIMINARY

Given a diagnostic dataset $\mathcal{D} = \{(C_i, y_i)\}_{i=1}^N$, where each conversation $C_i$ represents a dialogue between a patient and a healthcare provider, including both explicit symptoms (directly reported by the patient) and implicit symptoms (inferred from the dialogue context); $y_i \in \mathcal{Y}$ is the corresponding disease label; $N$ is the size of the dataset; the objective is to accurately predict $y_i$ based exclusively on the content of the dialogue. From each conversation $C_i$, a set of symptoms $\mathcal{S}_i = \{s_k\}_{k=1}^{|S_i|}$ is extracted. Each symptom $s_k$ is encoded into a high-dimensional semantic embedding using a pretrained text encoder, resulting in a matrix of symptom embeddings $\mathbf{e}^{(i)} \in \mathbb{R}^{|\mathcal{S}_i| \times d}$, where $d =$

3072 is the dimensionality of the embedding space. A fixed-size patient-level representation is then obtained by averaging the symptom embeddings:

$$\mathbf{E}_i = \frac{1}{|\mathcal{S}_i|} \sum_{k=1}^{|\mathcal{S}_i|} \mathbf{e}_k^{(i)} \tag{1}$$

While we use mean pooling to generate patient-level symptom embeddings for computational efficiency and vector normalization, this representation serves only as a first-stage input. The MPNN layer (Section 3.5) enhances this embedding by integrating neighborhood context via message passing and self-projection, reintroducing local granularity. Moreover, the LLM-based embeddings already encode symptom semantics, including severity indicators such as intensity adjectives (e.g., "mild" vs. "severe"), which are preserved in the vector space.

Our goal is to learn a function $\mathcal{F}$ that maps the dialogue $C_i$ (or equivalently, its extracted symptom representation $\mathbf{E}_i$) and its structural context in a patient-centric knowledge graph $\mathcal{G}$ to a predicted disease label $\hat{y}_i = \mathcal{F}(C_i, \mathcal{G})$. To this end, we formulate the learning objective as:

$$\mathcal{F}^* = \arg \min_{\mathcal{F} \in \mathcal{H}} \frac{1}{N} \sum_{i=1}^{N} \mathcal{L} \left( \mathcal{F}(C_i, \mathcal{G}), \, y_i \right) \tag{2}$$

where $\mathcal{H}$ denotes the space of candidate functions, and $\mathcal{L}(\cdot, \cdot)$ is the cross-entropy loss between the predicted and true disease labels. The function $\mathcal{F}$ should leverage both the semantic features captured in the dialogue and the graph-based relationships within $\mathcal{G}$ to enable accurate, interpretable, and structure-aware disease diagnosis.

### 3.2 OVERALL FRAMEWORK

The proposed SPKGDIAG combines LLMs with KGs reasoning for automated disease diagnosis (Figure 2). The model first extracts texts that describe symptoms using a GPT-based model to identify both explicit and implicit symptoms. These extracted symptom-related texts are converted into high-dimensional semantic vectors using OpenAI's text embedding model, with patient symptom embeddings combined into unified representations. A patient-centric knowledge graph, where nodes represent patients and edges connect patients with similar symptoms, is constructed to capture both individual and population-level clinical patterns. The graph is processed through a Graph Neural Network (GNN), specifically an MPNN, which allows information flow between neighboring nodes through multi-hop neighborhood sampling. The resulting node features are normalized and regularized, then passed through a feedforward neural network with softmax activation for diagnosis prediction. This work combines conversational understanding from language models with structured reasoning from graph networks, providing accurate and interpretable automated diagnosis by using both individual patient information and collective clinical knowledge.

### 3.3 SYMPTOM EXTRACTOR

SPKGDIAG first employs a symptom extractor using OpenAI's GPT-4.1[1] LLM, to automatically extract clinical symptoms from patient-provider dialogues, capitalizing on its advanced contextual understanding to process complex, unstructured conversational data. As illustrated in Figure 2, dialogue transcripts from multiple patients are input into Symptom Extractor, which identifies the text segments that describe the targeted symptoms, such as fever, cough, and headache, directly from the raw dialogue. The extraction process is guided by a medically structured prompt template (Appendix A.6) that ensures consistency, completeness, and clinical validity. Unlike generic summarization, this module is designed to identify both explicit and implicit symptoms with semantic granularity, enabling downstream modeling tasks such as graph construction and diagnosis prediction.

To obtain cross-symptom representations, we use the `text-embedding-3-large`[2] model to encode symptom-related text segments from multiple patients into high-dimensional vectors, capturing their semantic meaning and enabling the measurement of relatedness between different symptom

---

[1] https://openai.com/index/gpt-4-1/
[2] https://platform.openai.com/docs/models/text-embedding-3-large

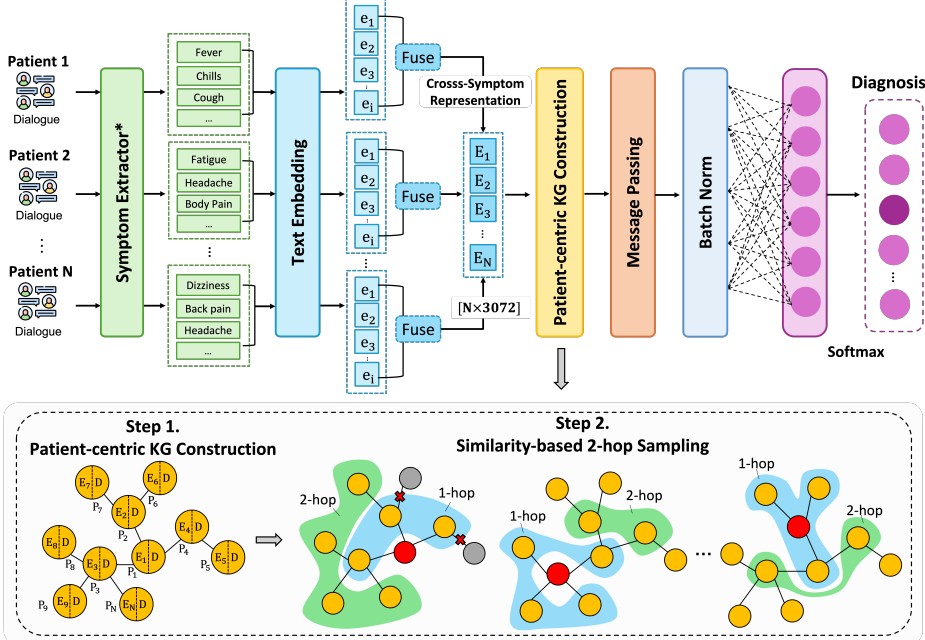

Figure 2: Architecture of the proposed SPKGDIAG framework for automated medical diagnosis. The model integrates LLMs with a patient-centric KG through a multi-stage pipeline. First, a symptom extractor identifies symptom-related text segments, which are transformed into semantic vector representations using text embeddings. Next, a patient-centric KG is constructed by connecting patients with similar symptom profiles, as depicted in the dotted region of the lower panel. A similarity-based 2-hop neighbor sampling strategy generates local subgraphs that capture extended patient relationships, which are processed by an MPNN layer with batch normalization. Finally, a softmax classifier produces diagnostic predictions.

texts. For patient $i$, the set of symptom embeddings is represented as $\mathbf{e}^{(i)} \in \mathbb{R}^{|\mathcal{S}_i| \times d}$. These symptom embeddings are then fused by computing their mean (Equation 1), resulting in a single vector $\mathbf{E}_i \in \mathbb{R}^d$ that encapsulates the overall semantic profile of patient $i$'s symptoms. This representation provides a unified, vectorized understanding of patient symptom profiles, which is particularly useful for tasks such as similarity retrieval and disease classification, leveraging the strengths of embeddings in search, clustering, and classification applications.

## 3.4 PATIENT-CENTRIC KNOWLEDGE GRAPH CONSTRUCTION

To construct a patient-centric knowledge graph, we leverage patient embeddings and graph topological structures to identify meaningful relationships among patient entities. Each node in the graph represents an individual patient and is enriched with both structural features and semantic information, denoted as $\mathbf{E}_i \in \mathbb{R}^d$ and its diagnosis label $\mathbf{D}$, respectively. Edges are established between patient nodes based on shared clinical symptoms, ensuring that the graph topology reflects clinically relevant associations such as common complaints, co-occurring presentations, or overlapping disease manifestations. This symptom-based connectivity facilitates the modeling of both explicit and latent clinical relationships across the patient population.

Formally, an adjacency matrix $\boldsymbol{A} \in \{0,1\}^{N \times N}$ is defined as follows:

$$A_{i,j} = A_{j,i} = \begin{cases} 1, & \text{if } \mathcal{S}_i \cap \mathcal{S}_j \neq \emptyset, \\ 0, & \text{otherwise.} \end{cases} \tag{3}$$

This definition guarantees that the graph is undirected and sparse, capturing only meaningful symptom-based patient connections. The corresponding graph $\mathcal{G} = (\mathcal{V}, \mathcal{E})$ comprises a vertex set $\mathcal{V}$, where each node $v_i \in \mathcal{V}$ represents a patient, and an edge set $\mathcal{E}$, where $(v_i, v_j) \in \mathcal{E}$ if and only if the entry $A_{i,j}$ in the adjacency matrix $\boldsymbol{A}$ is non-zero.

The construction process begins by computing a high-dimensional embedding matrix, where each row corresponds to a patient and captures semantic features derived from their dialogue or clinical data. To ensure that graph-based computations focus on meaningful clinical neighborhoods, pairwise similarities between patients are calculated using cosine similarity, but restricted to the local structure defined by $\mathbf{A}$. For patients $i$ and $j$, the cosine similarity is defined as:

$$\text{sim}(i, j) = \frac{\mathbf{E}_i \cdot \mathbf{E}_j}{\|\mathbf{E}_i\| \, \|\mathbf{E}_j\|}, \quad \text{only if } A_{i,j} = 1 \tag{4}$$

However, these comparisons are restricted to local neighborhoods as defined by the adjacency matrix $\mathbf{A}$, i.e., only for pairs $(i, j)$ where $A_{i,j} = 1$, This locality constraint preserves the sparsity and structural integrity of real-world healthcare data, which often exhibits naturally sparse connectivity due to varied diagnoses, treatment pathways, and healthcare encounters.

To further enrich the structural and semantic coherence of the graph, a multi-hop neighborhood sampling strategy is employed. For each patient node, the top-$k$ most similar neighbors are selected from its immediate (first-hop) connections based on cosine similarity:

$$\mathcal{N}_k(i) = \text{Top-}k\left(\{j \mid A_{i,j} = 1\}, \text{sim}(i, j)\right) \tag{5}$$

Subsequently, for each of the first-hop neighbors, an additional set of top-$k$ neighbors is sampled to form a second-hop neighborhood, excluding any nodes in the first-hop to minimize redundancy and encourage diversity:

$$\mathcal{N}_k^{(2)}(i) = \bigcup_{j \in \mathcal{N}_k(i)} \left(\mathcal{N}_k(j) \setminus (\mathcal{N}_k(i) \cup \{i\})\right) \tag{6}$$

We construct the final training graph by refining initial symptom-based edges through top-k cosine similarity sampling of patient embeddings (Equations 4, 5, and 6), filtering weak links. Avoiding complex, resource-heavy methods, we opt for a sparse, interpretable design that captures relevant patterns via localized sampling. This hierarchical strategy improves graph quality and clinical relevance, as shown in our ablation (Table 3).

The union of the seed node, its first-hop, and second-hop neighbors constitutes an expanded node set $\mathcal{V}_i = \{i\} \cup \mathcal{N}_k(i) \cup \mathcal{N}_k^{(2)}(i)$. From this, a sparse subgraph-specific adjacency matrix $\mathbf{A}^{(i)} \in \{0, 1\}^{|\mathcal{V}_i| \times |\mathcal{V}_i|}$ is reconstructed by preserving all edge relationships among the sampled nodes. This multi-hop neighborhood sampling procedure enables the construction of a contextually rich, patient-centered subgraph for each node. By capturing both local and extended patient similarities, the resulting graph effectively balances semantic richness and structural relevance.

### 3.5 PATIENT-CENTRIC MPNN DIAGNOSTIC

The framework employs a layered approach to learning node representations, using a message-passing neural network (MPNN) that combines message aggregation with a self-projection mechanism. This design is motivated by the need to balance semantic preservation with structural integration. The self-projection mechanism allows each node to retain its own symptom profile, while additive message aggregation incorporates clinically relevant patterns from neighboring patients. Together, these components enable each node to iteratively update its representation by integrating neighbor information while refining its intrinsic features. This approach is particularly effective in sparse, symptom-based graphs, as demonstrated in our ablation (Table 3, Figure 3), where the MPNN consistently outperforms GCN and GAT under identical settings. The effectiveness is further supported by the use of high-dimensional, LLM-derived embeddings that already encode rich semantic information.

The message-passing mechanism is realized through a transformation of the node features using a learnable weight matrix. For each edge in the graph, a message is computed and subsequently aggregated using an additive scheme. Let $x \in \mathbb{R}^{N \times d}$ denote the input node features, where $N$ is the number of nodes. Two trainable matrices $\mathbf{E}, \mathbf{T} \in \mathbb{R}^{d \times d}$ are employed for message transformation and self-projection, respectively. The messages $\mathbf{m}_i \in \mathbb{R}^d$ are calculated as: $\mathbf{m}_i = \sum_{j \in \mathcal{N}(i)} \text{norm}_{ij} \cdot (\mathbf{x}_j \mathbf{E})$, where $\mathcal{N}(i)$ denotes the set of neighbors of node $i$, and $\text{norm}_{i,j} = \frac{1}{\sqrt{\deg(i) \deg(j)}}$ serves as a symmetric normalization term derived from the degree of nodes, mitigating the impact of node

degree variability. Each node then updates its representation through a non-linear transformation that combines its self-projected features with the aggregated message. The update rule can be expressed as:

$$\mathbf{h}_i = \mathbf{x}_i + \sigma(\mathbf{x}_i \mathbf{T} + \mathbf{m}_i), \tag{7}$$

where $\sigma$ denotes a Leaky ReLU activation function, and $\mathbf{h}_i \in \mathbb{R}^d$ is the updated node embedding. To stabilize training and improve convergence, batch normalization is applied to the updated embeddings, followed by dropout regularization to prevent overfitting.

The overall architecture comprises a sequence of such graph convolutional layers, followed by a feedforward neural network for downstream tasks such as classification. The feedforward module includes a linear transformation to a hidden space, batch normalization, ReLU activation, and a final linear projection to the output space of class logits. Mathematically, the transformation can be described as: $\mathbf{z} = \text{ReLU}(\text{Dropout}(\text{BN}(\mathbf{h}\mathbf{W}_1)))\mathbf{W}_2$, where $\mathbf{W}_1 \in \mathbb{R}^{d \times d}$ and $\mathbf{W}_2 \in \mathbb{R}^{d \times c}$ are the learnable weight matrices of the linear layers, and $\mathbf{z} \in \mathbb{R}^c$ represents the final output logits per node.

This formulation supports both full forward propagation and partial forward propagation at a specific layer, which is useful for layer-wise analysis or interpretability in graph learning. The architecture is designed to balance expressivity and generalization, enabling it to effectively capture both local and global structural patterns in graph-based datasets.

## 4 EXPERIMENTS

### 4.1 EXPERIMENTAL SETTING

**Datasets.** The proposed approach was tested on four commonly utilized public datasets, such as MZ-4 (Wei et al., 2018), MZ-10 (Wei et al., 2018), Dxy (Xu et al., 2019), Synthetic (Liao et al., 2022), and an in-house VNPT dataset. A description of the datasets is provided in Appendix A.1.

**Baselines.** We evaluated our model against a number of baselines, including **ML models** (SVM (Chang & Lin, 2011)), **RL-based approaches** (PPO (Schulman et al., 2017), DQN (Wei et al., 2018)), **Non RL-based methods** (REFUEL (Peng et al., 2018), KR-DS (Xu et al., 2019), GAMP (Xia et al., 2020), HRL (Zhong et al., 2022) and BSODA (He et al., 2022)), **Transformer-based models** (Diaformer (Chen et al., 2022), DxFormer (Chen et al., 2023), CoAD (Wang et al., 2023) and MTDia (Hou et al., 2023)) and **Knowledge-enhanced and graph-based approaches** (Zhang et al. (2023), Tian et al. (2024), KDPoG (Li & Ruan, 2024) and EIRAD (Yan et al., 2024)). Details of the baselines and implementation settings are in Appendix A.2 and A.3, respectively.

### 4.2 COMPARISON PERFORMANCE

**Overall Performance.** Table 1 compares the performance of state-of-the-art diagnostic systems across four publicly available benchmark datasets using classification accuracy. We include results from prior studies when available; otherwise, we reproduce them using the official code, provided it is publicly accessible and executable. Our method outperforms all baselines, showing strong generalization from narrow (MZ-4, 4 diseases) to broad (Synthetic, 90 diseases) diagnostic tasks. Traditional machine learning models such as SVM-exp and SVM-exp&imp (Chang & Lin, 2011), which incorporate explicit and implicit symptoms, perform moderately (0.704 on MZ-4, 0.767 on Dxy) but struggle on complex datasets such as MZ-10 (0.633), due to limited symptom prediction and inability to model inter-patient correlations.

Several RL methods, such as DQN, REFUEL (Wei et al., 2018; Peng et al., 2018), HRL (Zhong et al., 2022), KR-DS (Xu et al., 2019), and GAMP (Xia et al., 2020), aim to simulate multi-round diagnostic inference through interactions. These models perform well on smaller datasets, achieving accuracies around 0.720-0.721 on Dxy. However, their performance degrades significantly on larger, noisier datasets like MZ-10, where DQN, for example, achieves only 0.408 accuracy. Their dependence on simulation environments and sparse reward signals leads to unstable training and limited generalization. While non-RL methods typically rely on transformers such as BSODA (He et al., 2022) that leverage knowledge-guided attention in a scalable non-RL context, achieving 0.802 and 0.747 accuracy on Dxy and Synthetic, respectively. Recent models such as DxFormer (Chen et al., 2023), Diaformer (Chen et al., 2022), and CoAD (Wang et al., 2023) improve symptom representation and disease prediction through deep contextual modeling. CoAD notably achieves 0.850

Table 1: Performance comparison across datasets using the accuracy metric. The best results are marked in bold, and the second-best results are marked with an underline. Entries marked with "–" indicate cases where neither comparable reported results nor runnable official code are available under our experimental setting.

| Method | MZ-4 | MZ-10 | Dxy | Synthetic |
|---|---|---|---|---|
| SVM-exp (Chang & Lin, 2011) | 0.685 | 0.547 | 0.621 | 0.341 |
| SVM-exp&imp (Chang & Lin, 2011) | 0.704 | 0.624 | 0.767 | 0.732 |
| PPO (Schulman et al., 2017) | 0.732 | – | 0.746 | 0.618 |
| DQN (Wei et al., 2018) | 0.690 | 0.408 | 0.720 | 0.356 |
| REFUEL (Peng et al., 2018) | 0.716 | 0.505 | 0.721 | – |
| KR-DS (Xu et al., 2019) | 0.730 | 0.485 | 0.740 | – |
| GAMP (Xia et al., 2020) | 0.730 | 0.500 | 0.769 | – |
| HRL (Zhong et al., 2022) | 0.694 | 0.556 | 0.695 | 0.496 |
| BSODA (He et al., 2022) | 0.731 | – | 0.802 | – |
| Diaformer (Chen et al., 2022) | 0.742 | – | 0.829 | 0.733 |
| DxFormer (Chen et al., 2023) | 0.743 | 0.633 | 0.817 | 0.712 |
| CoAD (Wang et al., 2023) | 0.750 | 0.628 | 0.850 | 0.727 |
| MTDiag (Hou et al., 2023) | 0.759 | – | 0.854 | 0.754 |
| Zhang et al. (2023) | 0.764 | – | 0.849 | 0.729 |
| KDPoG (Li & Ruan, 2024) | 0.754 | 0.568 | 0.837 | – |
| Tian et al. (2024) | 0.761 | – | 0.752 | – |
| EIRAD (Yan et al., 2024) | 0.768 | – | 0.845 | – |
| **SPKGDIAG** | **0.782** +1.4% | **0.677** +4.4% | **0.874** +2.0% | **0.828** +7.4% |

accuracy on Dxy. However, these models primarily focus on individual patients, limiting their ability to capture population-level symptom structures and broader clinical trends. In addition, MTDiag (Hou et al., 2023) addresses some of these limitations by integrating multi-task learning and LLM-based multi-expert reasoning, achieving 0.854 accuracy on Dxy. Nonetheless, it still lacks explicit mechanisms for modeling patient similarity or leveraging neighborhood structures in clinical data.

In contrast, recent graph-based approaches aim to address these limitations. KDPoG (Li & Ruan, 2024) captures heterogeneous patient connections, achieving 0.837 accuracy on Dxy, while Tian et al. (2024) employ a weighted heterogeneous knowledge graph for incremental, multi-department diagnosis, reaching 0.752 on Dxy. EIRAD (Yan et al., 2024) incorporates interpretable reasoning paths and evidence-aware rewards, achieving strong performance on both MZ-4 (0.768, second-best) and Dxy (0.845). In comparison, SPKGDIAG consistently outperforms all baselines across all datasets. These results underscore the strength of integrating patient-centric knowledge graphs with multi-hop neighborhood sampling, enabling robust, interpretable, and scalable diagnosis by capturing both individualized symptom profiles and population-level patterns. A case study and additional analyses are provided in Appendices A.4 and A.5, respectively.

Table 2: Performance comparison on the in-house dataset. The best results are marked in bold.

| Method | Accuracy | F1-score |
|---|---|---|
| Logistic Regression (Le et al., 2021) | 0.791 | 0.795 |
| DxFormer (Chen et al., 2023) | 0.793 | 0.796 |
| BiLSTM w/ Tokenizer (Nguyen et al., 2023) | 0.873 | 0.874 |
| SDCANet (Phan et al., 2023) | 0.883 | 0.881 |
| **SPKGDIAG** | **0.899** +1.6% | **0.898** +1.7% |

**Comparison Performance on the In-house Dataset.** As shown in Table 2, we compare our proposed method's performance against several existing approaches on a private VNPT dataset. SPKG-DIAG consistently outperformed all baseline models, achieving the highest accuracy of 0.899 and an F1-score of 0.898. These findings highlight the strong predictive performance and robustness of SPKGDIAG in modeling complex clinical data, thereby underscoring its potential for real-world healthcare applications.

## 4.3 ABLATION STUDY

### 4.3.1 IMPACT OF MODEL TYPE AND SIMILARITY-BASED $k$-HOP NEIGHBOR SAMPLING

Table 3: Ablation results on the similarity-based 2-hop sampling across different GNN variants

| Method | Similarity-based 2-hop Sampling | MZ-4 | MZ-10 | Dxy | Synthetic |
|---|---|---|---|---|---|
| SPKGDIAG$_{GCN}$ | ✗ | 0.634 | 0.460 | 0.612 | 0.483 |
| | ✓ | 0.662 | 0.543 | 0.728 | 0.703 |
| SPKGDIAG$_{GAT}$ | ✗ | 0.697 | 0.565 | 0.738 | 0.519 |
| | ✓ | 0.754 | 0.604 | 0.806 | 0.787 |
| SPKGDIAG | ✗ | 0.761 | 0.625 | 0.835 | 0.801 |
| | ✓ | **0.782** | **0.677** | **0.874** | **0.823** |

Table 3 demonstrates that incorporating similarity-based 2-hop neighbor sampling consistently enhances diagnostic performance across all SPKGDIAG variants and datasets. Notably, SPKG-DIAG$_{GCN}$ — which integrates Graph Convolutional Networks (GCN) (Kipf & Welling, 2017) and SPKGDIAG$_{GAT}$ — which adopts Graph Attention Networks (GAT) (Veličković et al., 2018), both benefit from this architectural enhancement. For instance, SPKGDIAG$_{GCN}$ improves from 0.634 to 0.662 on MZ-4, and more dramatically from 0.483 to 0.703 on the Synthetic dataset. Similarly, SP-KGDIAG$_{GAT}$ improves from 0.697 to 0.754 on MZ-4, and from 0.519 to 0.787 on Synthetic. These improvements are even more striking in the base SPKGDIAG model, which adopts an MPNN architecture. With 2-hop sampling, it achieves the highest and most consistent gains across all datasets. Conversely, the absence of similarity-based 2-hop sampling results in notable performance drops, particularly on the Synthetic dataset. Here, SPKGDIAG$_{GCN}$ drops by over 22% (from 0.703 to 0.483), and SPKGDIAG$_{GAT}$ by more than 26% (from 0.787 to 0.519), indicating that strictly local aggregation fails to capture sufficient structural context.

Overall, these results clearly demonstrate that similarity-based 2-hop neighbor sampling is a robust and scalable architectural enhancement. Expanding the receptive field enables GNNs to capture richer structural and semantic information from the knowledge graph, leading to significantly more accurate diagnostic predictions across diverse architectures and datasets.

### 4.3.2 IMPACT OF GRAPH, SELF-PROJECTION, AND MESSAGE AGGREGATION IN SPKGDIAG

Figure 3 presents an ablation study assessing the contributions of the Self-Projection mechanism, message aggregation, and graph structure in the MPNN-based SPKGDIAG framework. Removing Self-Projection consistently leads to noticeable drops in accuracy across all datasets, highlighting its role in maintaining stable and expressive node representations. For instance, on MZ-4, accuracy decreases from 0.782 to 0.725 without Self-Projection, compared to 0.732 without message aggregation. A similar pattern appears on MZ-10, where the full model achieves 0.672, while the ablated variants yield 0.658 and 0.668. On the Synthetic dataset, the model reaches 0.828, outperforming the ablations at 0.802 and 0.810. On Dxy, message aggregation has a slightly larger effect, dropping accuracy from 0.874 to 0.816, while removing Self-Projection reduces it to 0.864. The most significant performance drop occurs when the graph structure is removed, indicated as "w/o Graph", which eliminates both

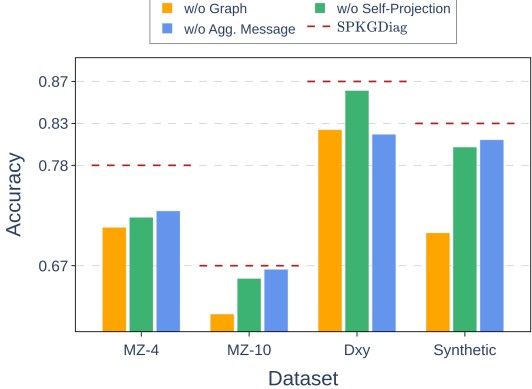

Figure 3: Ablation study demonstrating the effectiveness of SPKGDiag with MPNN's Self-Projection mechanism. "Agg. Message" refers to the aggregated message in MPNN. Here, SPKG-DIAG denotes our fully implemented model.

the "Patient-centric KG Construction" and "Message Passing" components, while preserving all other modules, as shown in Figure 2. In this setting, accuracy drops to 0.714 on MZ-4, 0.619 on MZ-10, 0.821 on Dxy, and 0.708 on Synthetic, consistently the lowest across all variants. These findings confirm that while Self-Projection and message aggregation support effective node-level learning, the graph structure is essential for relational reasoning and information propagation. Overall, high diagnostic performance in SPKGDIAG relies on the integration of all three components: graph connectivity, message aggregation, and Self-Projection.

### 4.3.3    IMPACT OF NEIGHBORHOOD DEPTH IN $k$-HOP NEIGHBOR SAMPLING

Figure 4 illustrates that 2-hop neighbor sampling consistently yields the highest accuracy across most datasets, highlighting its effectiveness in capturing clinically meaningful relationships. Specifically, transitioning from 1-hop to 2-hop neighborhoods results in notable performance gains (+2.8% on MZ-4 and +1.0% on Dxy). This suggests that considering patients with similar but not necessarily identical symptom profiles enhances diagnostic reasoning, which aligns with real-world clinical practices where physicians factor in related cases to inform differential diagnoses. In contrast, the consistent performance decline observed with 3-hop sampling (–2.0% on Dxy and –1.4% on MZ-4) indicates that expanding the neighborhood too far introduces noise from distantly connected and weakly correlated patients. This highlights a trade-off in neighborhood selection, where broader context may become less clinically meaningful and potentially misleading. Interestingly, the Synthetic dataset shows minimal variation across hop sizes. This implies that real-world clinical data, which contain complex comorbidity structures and heterogeneous symptom presentations, benefit more from multi-hop reasoning than simplified synthetic data is able to reveal.

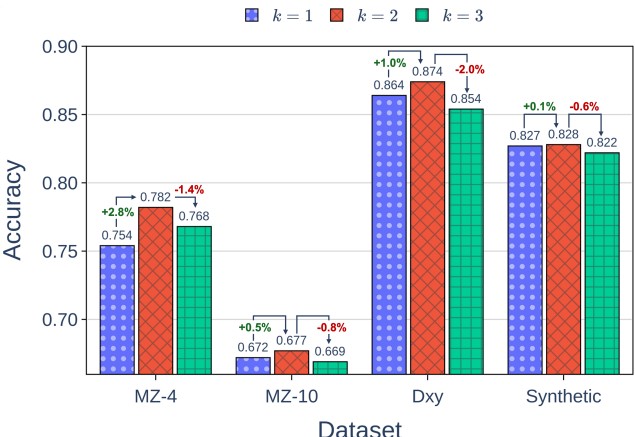

Figure 4: Ablation results on neighborhood depth in $k$-hop sampling for the SPKGDIAG model

## 5    CONCLUSION

In this study, we presented SPKGDIAG, a novel framework that combines large language models with patient-centered knowledge graphs to improve the accuracy and interpretability of automated disease diagnosis. Our method used LLM to extract explicit and implicit symptoms from patient-doctor conversations, allowing a more comprehensive understanding of clinical presentations. By constructing a symptom-based knowledge graph and using MPNN with similarity-based multi-hop neighbor sampling, the framework was able to capture both local individual-level and population-level patient representations. Our framework offers a pragmatic alternative to resource-heavy or ontology-dependent systems and is particularly suitable for deployment where structured medical data or multimodal alignment is unavailable. Extensive experimental results across four public datasets demonstrated that our approach significantly outperformed the state-of-the-art performance, achieving an improvement in diagnostic accuracy of up to 7.4%. For future work, we plan to incorporate dynamic graph construction for evolving patient interactions, model temporal and longitudinal clinical data to capture disease progression, and explore methods for encoding symptom severity beyond mean pooling. Additionally, we aim to enhance scalability through subgraph caching and efficient neighborhood retrieval to support deployment in large-scale healthcare systems.

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

# A APPENDIX

## A.1 DATASETS

We tested on four commonly utilized public datasets, such as MZ-4, MZ-10, Dxy, Synthetic, and an in-house VNPT dataset. Table 4 provides a comparison of these datasets in terms of the number of diseases, symptoms, sizes of training/test sets, as well as the average degree and density of their symptom-disease graphs. The structure of each symptom-disease graph is quantitatively described using two key metrics: *average degree* and *density*. The **average degree** reflects the average number of edges connected to a node and provides insight into how interconnected the diseases and symptoms are within the graph. In contrast, **density** measures the proportion of actual edges to the maximum possible number of edges in the graph. These metrics are formally defined as:

$$\begin{cases} \text{Average Degree} = \dfrac{1}{|V|} \sum_{u \in V} \deg(u) \\ \text{Density} = \dfrac{2 \cdot |E|}{|V| \cdot (|V| - 1)} \end{cases} \tag{8}$$

where $|V|$ denotes the number of nodes and $|E|$ the number of undirected edges. A higher density indicates a more fully connected graph, while a lower density suggests sparsity, which is common in real-world medical knowledge graphs due to the selective symptom-disease associations.

Table 4: Comparison of datasets based on diseases, symptoms, training, and test samples. "#" denotes "the number of". "Avg. Degree" and "Std. Degree" refer to the average and standard deviation of the degree (i.e., number of connections) per patient in the graph. "Avg. Disease" and "Std. Disease" represent the average and standard deviation of disease distribution across patients. An asterisk (*) indicates the inclusion of both actual symptoms and unrelated words automatically extracted by OpenAI's symptom extractor module.

| | MZ-4 | | MZ-10 | | Dxy | | Synthetic | | VNPT | |
|---|---|---|---|---|---|---|---|---|---|---|
| | Train | Test | Train | Test | Train | Test | Train | Test | Train | Test |
| # Samples | 568 | 142 | 3,305 | 811 | 423 | 104 | 24,000 | 6,000 | 184,383 | 46,096 |
| # Density | 0.267 | 0.270 | 0.270 | 0.270 | 0.333 | 0.321 | 0.054 | 0.053 | 0.033 | 0.034 |
| Avg. Degree | 151.51 | 38.07 | 890.96 | 218.00 | 139.95 | 32.74 | 1282.89 | 318.99 | 3602.71 | 1565.47 |
| Std. Degree | 100.91 | 24.93 | 655.71 | 163.67 | 90.50 | 22.05 | 1094.11 | 272.18 | 5303.53 | 2286.41 |
| Avg. Disease | 142.00 | 35.50 | 330.50 | 81.10 | 84.60 | 20.80 | 266.67 | 66.67 | 2832.30 | 827.90 |
| Std. Disease | 22.46 | 6.56 | 106.52 | 27.69 | 7.54 | 1.79 | 17.88 | 7.81 | 2065.45 | 604.22 |
| # Diseases | 4 | | 10 | | 5 | | 90 | | 10 | |
| # Symptoms | 66 | | 331 | | 41 | | 266 | | 36,588* | |

- **MZ dataset** (Wei et al., 2018), from the Pediatric Department of Baidu Muzhi[3] for the evaluation of automatic diagnostic systems, contains 710 user objectives and 66 symptoms for four categories of disorders (children's bronchitis, functional dyspepsia, infantile diarrhea infection, and upper respiratory infection). MZ-10 is a multi-level annotated dataset expanded from MZ-4 to include 10 diseases, encompassing common respiratory, endocrine, and digestive disorders, as well as a broader set of annotated symptoms.

- **Dxy dataset** (Xu et al., 2019) is an annotated medical conversation dataset obtained from Dingxiang Doctor[4], a popular Chinese online healthcare service. It contains 527 user objectives and 41 symptoms across 5 categories of disorders (allergic rhinitis, upper respiratory infection, pneumonia, children's hand-foot-mouth disease, and pediatric diarrhea).

- **Synthetic dataset** (Liao et al., 2022) is based on SymCat2, a database of symptom-related diseases. It has 30,000 user objectives and 90 illnesses.

- **VNPT dataset** is an in-house, large-scale real-world clinical dataset comprising 230,479 patient records collected from March 2016 to March 2021 at the Medical Center of My Tho City, Tien Giang Province, Vietnam. It includes patient-reported symptoms and diagnoses across 10 common disease categories, with respiratory, endocrine, and circulatory system disorders being the most prevalent (as detailed in Table 5). The dataset offers a diverse and realistic setting for automated medical diagnosis. The symptom set features over 36,000 unique terms, encompassing both actual symptoms and unrelated words automatically extracted by OpenAI's symptom extractor module, some of which may not directly correspond to clinical symptom expressions.

Figure 5 presents the class distribution of four datasets containing ten or fewer categories. The MZ-4, MZ-10, and Dxy datasets display relatively balanced distributions, with each class contributing a similar proportion of samples. In contrast, the VNPT dataset shows a noticeable level of class imbalance, as several classes occupy only a very small percentage of the total data. This comparison highlights that most datasets are well balanced, while VNPT requires special consideration due to its uneven class representation.

---

[3] https://muzhi.baidu.com/
[4] https://dxy.com/

Table 5: Distribution of disease categories in the VNPT dataset used for automated diagnosis.

| No. | Disease Name | #Samples |
|---|---|---|
| 1 | Respiratory System Diseases | 41,888 |
| 2 | Endocrine, Nutritional & Metabolic Disorders | 38,672 |
| 3 | Circulatory System Diseases | 37,782 |
| 4 | Musculoskeletal & Connective Tissue Diseases | 35,427 |
| 5 | Eye & Adnexa Diseases | 18,443 |
| 6 | Genitourinary System Diseases (B212) | 17,503 |
| 7 | Neoplasms | 16,271 |
| 8 | Injury, Poisoning & External Causes | 13,783 |
| 9 | Skin & Subcutaneous Tissue Diseases | 7,044 |
| 10 | Pregnancy, Childbirth & Puerperium | 3,666 |

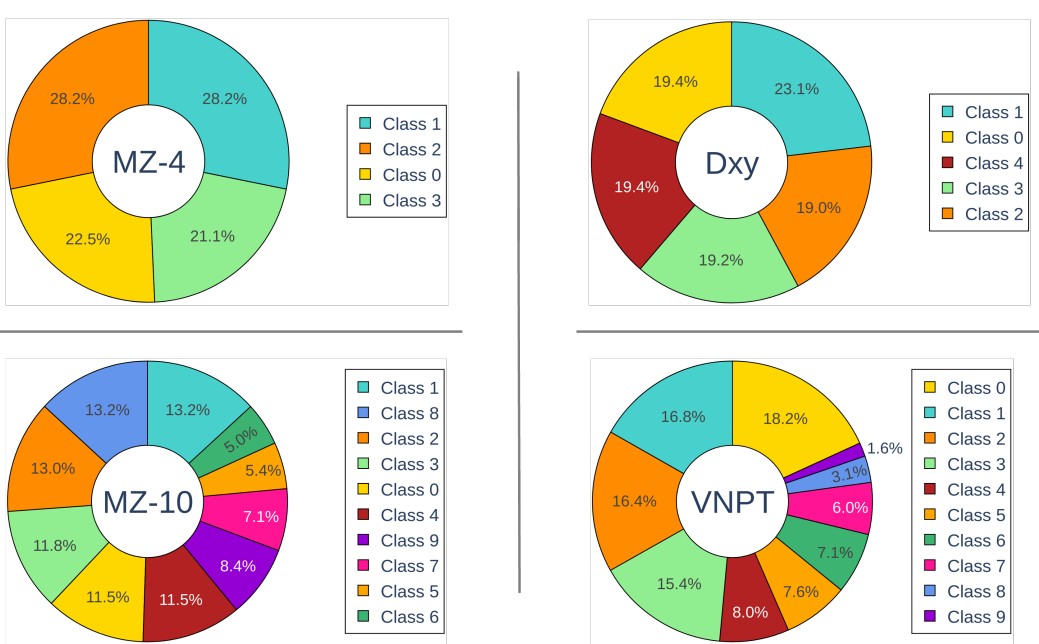

Figure 5: Class distribution visualization across datasets with 10 or fewer categories

To prevent any risk of information leakage, we first split each dataset into training and test sets. We then construct separate patient-centric graphs for training and testing. The training graph is built using only the training set, and is used to learn model parameters. For evaluation, a test graph is independently constructed using only test set patients and their symptoms. No edges or nodes are shared across splits, ensuring a strict separation between training and inference stages.

## A.2 BASELINES

We evaluated our model against a range of baseline approaches, spanning both conventional and state-of-the-art methods:

- **SVM** (Chang & Lin, 2011): A non-interactive method that utilizes both explicit and implicit symptoms to build a strong feature-based classifier.

- **RL-based methods: PPO** (Schulman et al., 2017), **DQN** (Wei et al., 2018): Standard RL-based models simulating symptom acquisition and decision-making.

- **Non RL-based methods: REFUEL** (Peng et al., 2018), **KR-DS** (Xu et al., 2019), **GAMP** (Xia et al., 2020), **HRL** (Zhong et al., 2022): Enhanced RL variants employing adversarial training, hierarchical structures, reward shaping, or knowledge graphs. **BSODA** (He et al., 2022): A scalable non-RL method using knowledge-guided attention mechanisms.

- **Transformer-based models: Diaformer** (Chen et al., 2022), **DxFormer** (Chen et al., 2023), **CoAD** (Wang et al., 2023) decouple or jointly model symptom inquiry and diagnosis using sequence modeling and label expansion to improve diagnostic accuracy. **MTDiag** (Hou et al., 2023): Replaces RL with multi-task classification and contrastive learning.

- **LLM-integrated models:** Incorporate LLMs with experiential medical knowledge (Zhang et al., 2023).

- **Graph-based models: KDPoG** (Li & Ruan, 2024), Tian et al. Tian et al. (2024), and **EIRAD** (Yan et al., 2024) leverage heterogeneous medical graphs for structured reasoning and knowledge integration.

## A.3 IMPLEMENTATION DETAILS

In our implementation, we adopted a configurable MPNN framework developed in Python, leveraging PyTorch for general deep learning operations and PyTorch Geometric (PyG) (Fey & Lenssen, 2019) for efficient graph representation learning. Input symptoms are embedded using OpenAI's `text-embedding-3-large` model. The hidden node feature dimension is set to 100, and the model operates over a 2-hop neighborhood, sampling 8 neighbors per hop to capture both immediate and extended patient similarities. Given the sparsity of the constructed graph and its emphasis on local structure, we employed a single message-passing layer with element-wise addition for message aggregation, offering a balance between simplicity and effectiveness in sparse settings. To mitigate overfitting, we applied a dropout rate of 0.42 after aggregation and used batch normalization to stabilize training and improve convergence. The model was optimized using the Adagrad optimizer with a learning rate of 6e-4, combined with a cosine annealing learning rate scheduler. Training was performed with a batch size of 8 over 50 epochs on a workstation equipped with an NVIDIA RTX A5000 (24 GB) GPU and an AMD EPYC 7302 16-core processor.

We recognize the importance of scalability for real-world deployment. To this end, our method maintains computational efficiency via several design choices: 1) the graph is sparse by design (Equation 3), 2) neighborhood sampling is restricted to 2-hop local subgraphs, ensuring memory efficiency, 3) training operates in mini-batches using PyTorch Geometric's efficient sparse matrix operations. This design allows us to scale the model efficiently without materializing the full adjacency matrix, while preserving clinical relevance through symptom-based connectivity.

## A.4 CASE STUDY AND INTERPRETABILITY ANALYSIS: MPNN EFFECTIVENESS IN NOISY GRAPH STRUCTURES

This analysis focuses on Test Node 79 (Patient ID: 79) to illustrate the interpretability and performance of the SPKGDIAG when applied to a densely connected and noise-prone graph structure. Patient 79 is drawn from the MZ-4 dataset, which comprises four pediatric disease categories: 0 - Upper respiratory infection, 1 - Pediatric bronchitis, 2 - Pediatric diarrhea, and 3 - Pediatric dyspepsia. Graph edges are defined by the presence of at least one shared symptom between patients, a construction rule that intentionally amplifies structural noise. In this case, the subject exhibits a highly mixed symptom profile that overlaps with multiple disease classes. Despite this, our proposed model, SPKGDIAG, successfully classified Patient 79 as Pediatric bronchitis (Label 1), which aligns with the ground-truth diagnosis. As illustrated in Figure 6, the model navigates a structurally ambiguous region where strong signals originate from multiple competing classes. The visualization highlights the pathway of the strongest signals, demonstrating SPKGDIAG's robustness and its ability to learn discriminative patterns from noisy and overlapping feature spaces.

### A.4.1 QUANTITATIVE INTERPRETABILITY: SIMILARITY VS. INFLUENCE

**Analysis of Cosine Similarity (Structural Proximity)**

The first-hop neighbors of Node 79 steadily exhibit high cosine similarity values, reflecting the density and strong structural coherence introduced by the permissive graph rule. As shown in Table 6, Node 37 (with Label 2) records the highest cosine similarity score of 0.9027 among these neighbors.

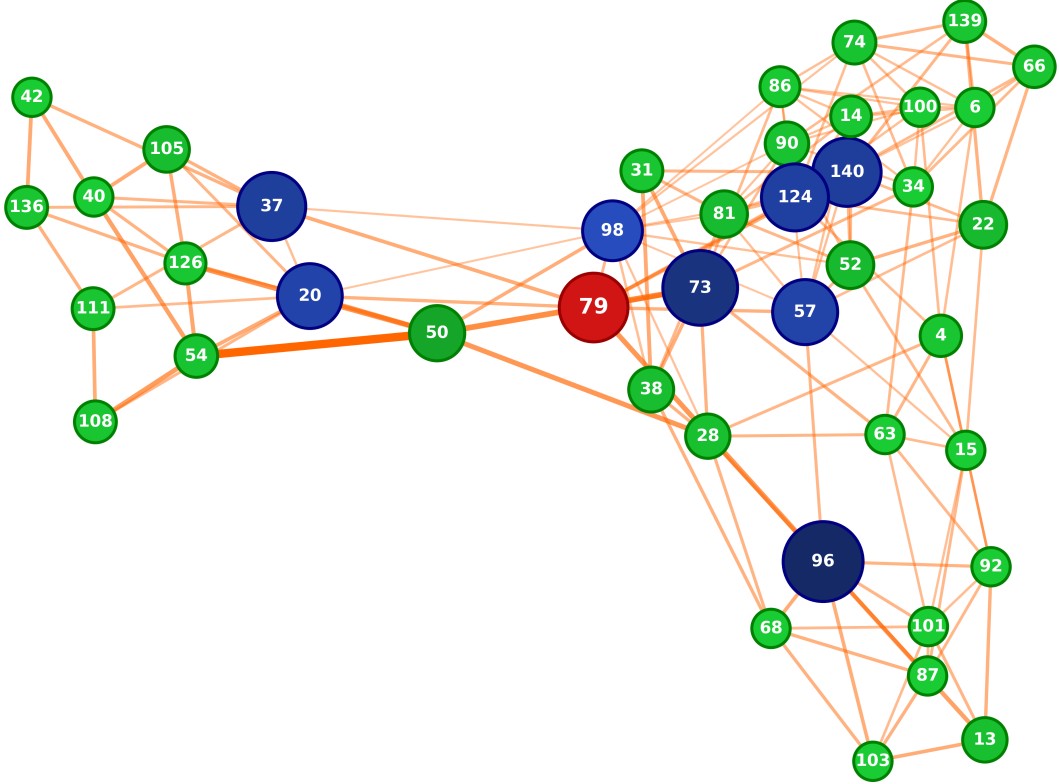

Figure 6: Graph-based visualization of influence propagation centered on Node 79. Node size and color intensity represent Message Norm; edge thickness and color indicate connection strength. Blue nodes are first-hop neighbors, green nodes are second-hop neighbors. Node 79 is centrally located between symptom clusters corresponding to respiratory diseases (Labels 0 and 1) and gastrointestinal diseases (Labels 2 and 3), receiving strong influence from diverse neighbors such as Node 96 (Label 1), Node 73 (Label 0), and Node 37 (Label 2). This structure illustrates how the SPKGDIAG leverages local similarity while filtering competing class signals in a complex topology.

Table 6: Cosine similarity scores of source nodes with respect to the target node 79

| Source Node | Label | Cosine Similarity |
|---|---|---|
| 37 | 2 | **0.9027** |
| 57 | 1 | 0.8968 |
| 98 | 1 | 0.8929 |
| 20 | 3 | 0.8861 |
| 140 | 1 | 0.8758 |
| 124 | 0 | 0.8729 |

However, Node 37 belongs to a different class, Pediatric diarrhea, which competes with the true label of Node 79. This finding highlights a critical limitation: relying solely on local feature similarity for classification may lead to influence from highly similar yet semantically irrelevant neighbors.

**Message Norm Analysis (Model Influence)**

The model's effectiveness is demonstrated by its ability to modulate incoming message weights, prioritizing diagnostic relevance over structural similarity within the graph. As illustrated in Figure 7 and detailed in Table 7, Node 79 received substantial adversarial input from neighboring nodes associated with competing classes. These include the Upper Respiratory class (Label 0), notably Node 73 (0.3506) and Node 124 (0.2569), linked through shared symptoms such as Sputum and Cough; and Gastrointestinal (GI) classes (Labels 2 and 3), including Node 37 (Label 2, 0.2700)

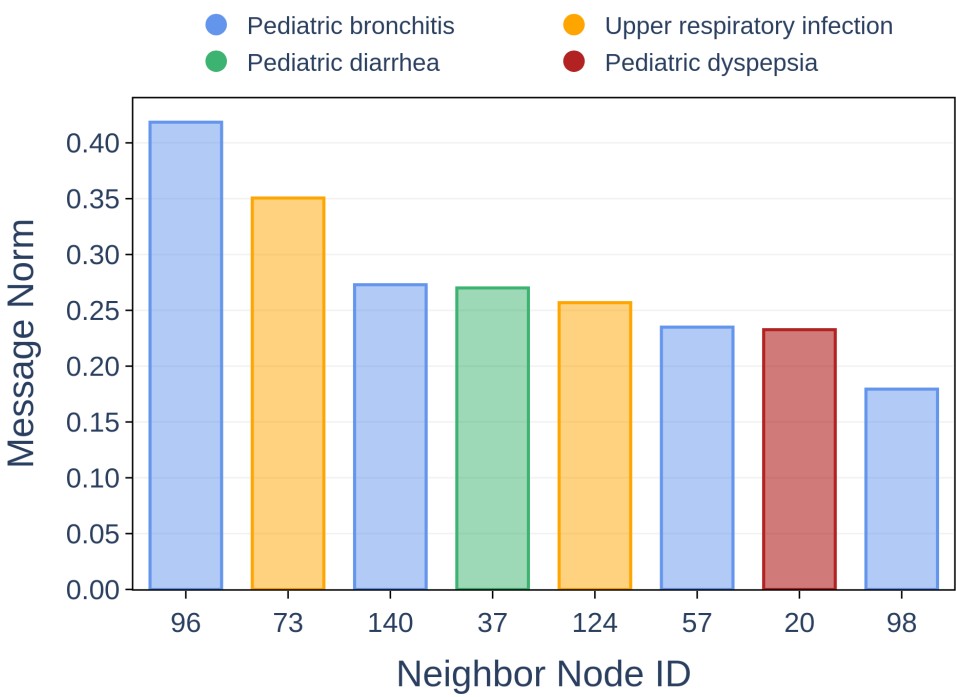

Figure 7: Message norms of top neighbor nodes by diagnosis category

and Node 20 (Label 3, 0.2326), with overlapping features like Dyspepsia and Cough. Despite these strong but non-specific connections, Node 96 (Label 1) – associated with the correct class, Pediatric bronchitis – produced the highest message norm (0.4185). This occurred even though its cosine similarity (0.8687) was lower than that of Node 37 (0.9027), underscoring the SPKGDIAG's ability to prioritize clinically salient features over superficial symptom-level similarity.

Table 7: Shared symptoms and rationale for links between Node 79 and its neighbors

| Neighbor (Label) | Shared Symptoms | Rationale for Link |
|---|---|---|
| Node 96 (L = 1) | Sputum, Wheezing, Cough | Strong feature overlap, reinforcing the correct class (Pediatric bronchitis). |
| Node 73 (L = 0) | Sputum, Cough | Connection via basic common respiratory symptoms, generating a strong adversarial signal (0.3506). |
| Node 37 (L = 2) | Sputum, Wheezing, Dyspepsia | Cross-class linkage via respiratory symptoms (Sputum, Wheezing) and GI symptoms (Dyspepsia). |
| Node 20 (L = 3) | Cough, Dyspepsia | Linkage via the most common respiratory symptom (Cough) and digestive features, despite being Label 3. |

Moreover, second-hop neighbors exhibited minimal message norms (e.g., Node 4: 0.0203; Node 6: 0.0000; Node 136: 0.0239), highlighting their relatively weak contribution compared with high-relevance first-hop nodes. This further confirms that SPKGDIAG selectively emphasizes localized, semantically rich interactions over noisy or distant connections. Overall, the model's behavior reflects a structured and clinically coherent reasoning process, successfully down-weighting noisy adversarial signals while reinforcing features indicative of the correct class.

### A.4.2 MESSAGE INFLUENCE HEATMAP

Figure 8 presents a detailed heatmap of normalized message norms, capturing the strength of communication from source to destination nodes within a network. The intensity of each cell reflects the magnitude of the message transmitted, with higher values (toward yellow) indicating stronger influence. The overall pattern reveals a sparse and uneven distribution of communication, where the majority of node pairs exchange minimal information. Notably, nodes 50 and 54 exhibit prominently high message norms directed toward target node 79, suggesting a concentrated influence on this particular destination. This indicates that node 79 is selectively integrating information from a small subset of source nodes, rather than uniformly across the network.

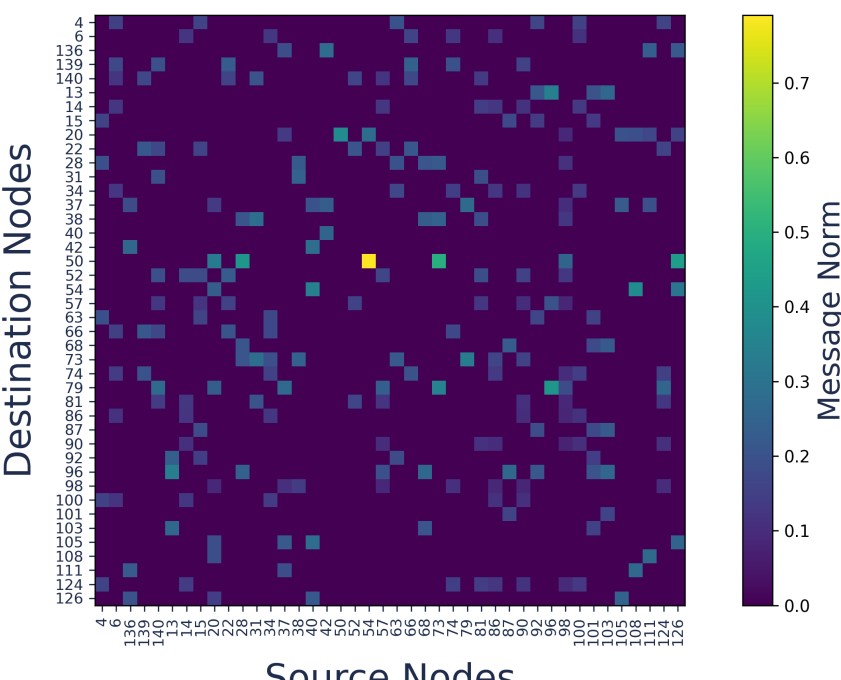

Figure 8: Message influence heatmap of normalized communication strengths between nodes

### A.5 FURTHER ANALYSIS

### A.5.1 CLASSIFICATION PERFORMANCE METRICS FOR MZ-4, MZ-10, AND DXY DATASETS

To ensure a clinically meaningful evaluation of model performance, we report a comprehensive set of metrics beyond overall accuracy, including per-class Precision, Recall, F1-score, PR-AUC (Precision-Recall Area Under the Curve), and Support, along with macro and weighted averages for each dataset (see Table 8). These metrics capture not only overall accuracy but also how well the model detects both common and rare conditions. Precision reflects the reliability of positive predictions, Recall captures sensitivity to true cases, and PR-AUC summarizes the balance between them, particularly under class imbalance. In the MZ-10 dataset, the model performs well on clinically important but less prevalent classes such as class 5 and class 7, with high Precision (0.973, 0.966), Recall (0.800, 0.966), F1-score (0.878, 0.966), and PR-AUC (0.925, 0.965), indicating accurate and sensitive predictions for these high-risk categories. In contrast, lower performance on classes such as class 3 and class 9 highlights limitations in detecting some rarer conditions. Similar trends are seen in the Dxy dataset, where class 2 underperforms (F1-score: 0.588, PR-AUC: 0.703) compared to consistently high scores in classes 0, 3, and 4. These results underscore the importance of detailed, class-specific evaluation to uncover both the strengths and failure modes of the model, ensuring its reliability across both common and rare diagnostic categories.

Table 8: Detailed classification performance metrics for MZ-4, MZ-10 and Dxy dataset

| Dataset | Class | Precision | Recall | F1-score | PR-AUC | Support |
|---|---|---|---|---|---|---|
| **MZ-4** | 0 | 0.846 | 0.733 | 0.786 | 0.781 | 30 |
| | 1 | 0.763 | 0.853 | 0.806 | 0.815 | 34 |
| | 2 | 0.750 | 0.933 | 0.832 | 0.812 | 45 |
| | 3 | 0.818 | 0.545 | 0.655 | 0.633 | 33 |
| | macro avg | 0.794 | 0.766 | 0.769 | 0.760 | 142 |
| | weighted avg | 0.789 | 0.782 | 0.775 | 0.764 | 142 |
| **MZ-10** | 0 | 0.637 | 0.691 | 0.663 | 0.704 | 94 |
| | 1 | 0.752 | 0.752 | 0.752 | 0.755 | 109 |
| | 2 | 0.705 | 0.692 | 0.698 | 0.745 | 107 |
| | 3 | 0.698 | 0.381 | 0.493 | 0.481 | 97 |
| | 4 | 0.521 | 0.526 | 0.524 | 0.551 | 95 |
| | 5 | 0.973 | 0.800 | 0.878 | 0.925 | 45 |
| | 6 | 0.733 | 0.717 | 0.725 | 0.713 | 46 |
| | 7 | 0.966 | 0.966 | 0.966 | 0.965 | 58 |
| | 8 | 0.674 | 0.835 | 0.746 | 0.727 | 109 |
| | 9 | 0.343 | 0.480 | 0.400 | 0.368 | 50 |
| | macro avg | 0.700 | 0.684 | 0.685 | 0.693 | 810 |
| | weighted avg | 0.690 | 0.677 | 0.675 | 0.685 | 810 |
| **Dxy** | 0 | 0.944 | 0.850 | 0.895 | 0.944 | 20 |
| | 1 | 0.594 | 0.826 | 0.691 | 0.726 | 23 |
| | 2 | 0.714 | 0.500 | 0.588 | 0.703 | 20 |
| | 3 | 0.857 | 0.900 | 0.878 | 0.966 | 20 |
| | 4 | 0.944 | 0.850 | 0.895 | 0.971 | 20 |
| | macro avg | 0.811 | 0.785 | 0.789 | 0.862 | 103 |
| | weighted avg | 0.804 | 0.786 | 0.786 | 0.858 | 103 |

### A.5.2 CONFUSION MATRIX FOR DIAGNOSTIC PERFORMANCE

We present the normalized confusion matrix for SPKGDIAG's diagnostic performance on the MZ-4, MZ-10, and Dxy dataset. Overall, SPKGDIAG successfully identified and differentiated features derived from both explicit and implicit symptoms. On the MZ-4 dataset (Figure 9), the model performs well on pediatric diarrhea (0.93) and bronchitis (0.82), though pediatric dyspepsia shows some confusion with diarrhea, suggesting these conditions share similar clinical features. The MZ-10 results (Figure 10) illustrate consistent performance across ten different conditions, with neonatal jaundice (0.97) and pediatric fever (0.83) achieving the highest accuracy rates, while some respiratory diseases show overlapping predictions due to their comparable symptoms. The Dxy dataset (Figure 11) confirms the model's ability to achieve nearly perfect classification, with pediatric diarrhea reaching complete accuracy (1.00) and hand-foot-mouth disease showing minimal errors (0.95), proving the system's effectiveness in distinguishing between different pediatric medical conditions.

### A.5.3 T-SNE VISUALIZATION OF PATIENT EMBEDDING REPRESENTATIONS

The t-SNE visualizations of learned patient embeddings across three datasets illustrate the model's effectiveness in creating meaningful diagnostic representations within a two-dimensional space. Figure 12 (MZ-4) shows clearly separated clusters for four pediatric conditions, where pediatric bronchitis, upper respiratory infection, and pediatric diarrhea form distinct groups, although some overlap between pediatric dyspepsia and diarrhea indicates their similar gastrointestinal symptoms. Figure 13 (MZ-10) presents more complex clustering arrangements across ten conditions, with certain diseases such as neonatal jaundice and pediatric constipation forming well-defined, separate clusters, while respiratory conditions appear closer together due to their comparable clinical features. Figure 14 (Dxy) demonstrates excellent cluster separation across five conditions, with each disease category occupying different areas of the embedding space, particularly hand-foot-mouth disease and allergic rhinitis showing complete separation, which confirms the model's ability to identify clinically significant diagnostic differences and supports the high classification performance shown in the confusion matrices.

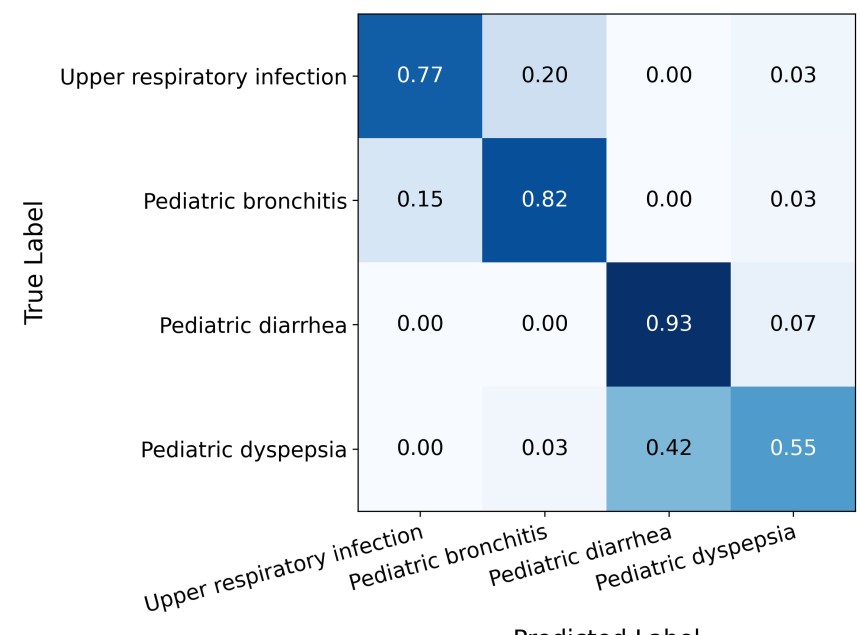

Figure 9: Confusion matrix for diagnostic performance on the Muzhi-4 dataset

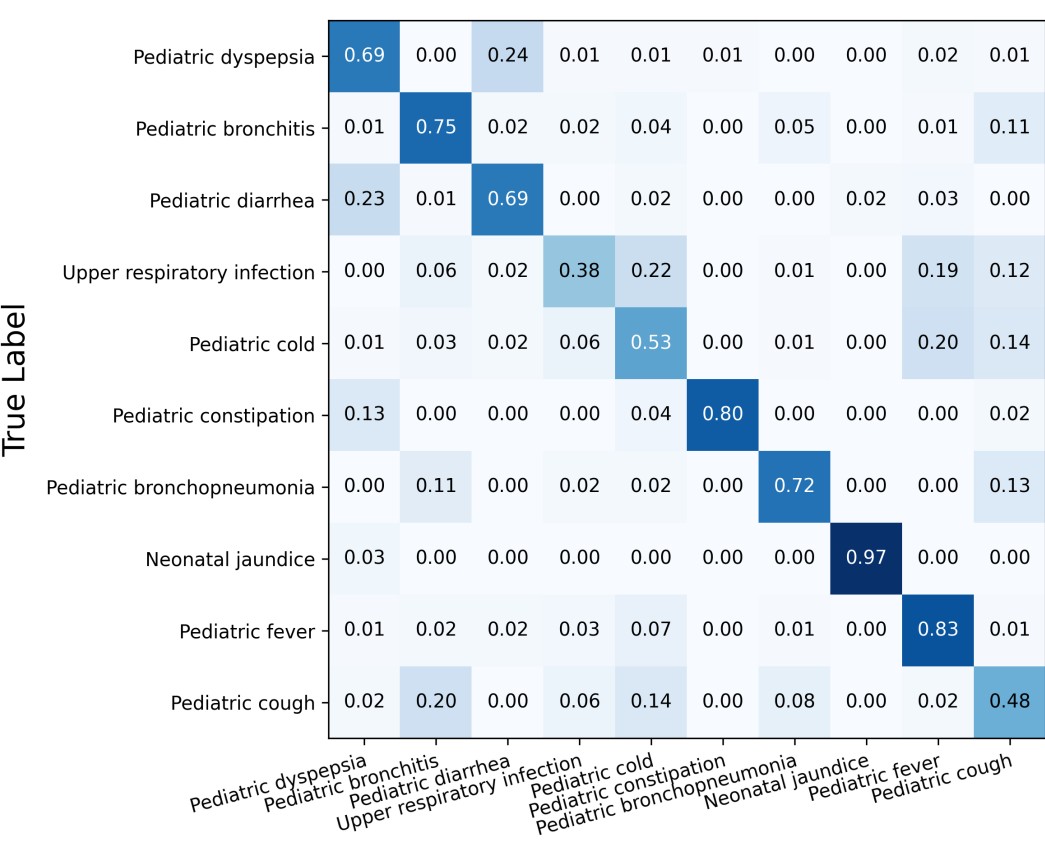

Figure 10: Confusion matrix for diagnostic performance on the Muzhi-10 dataset

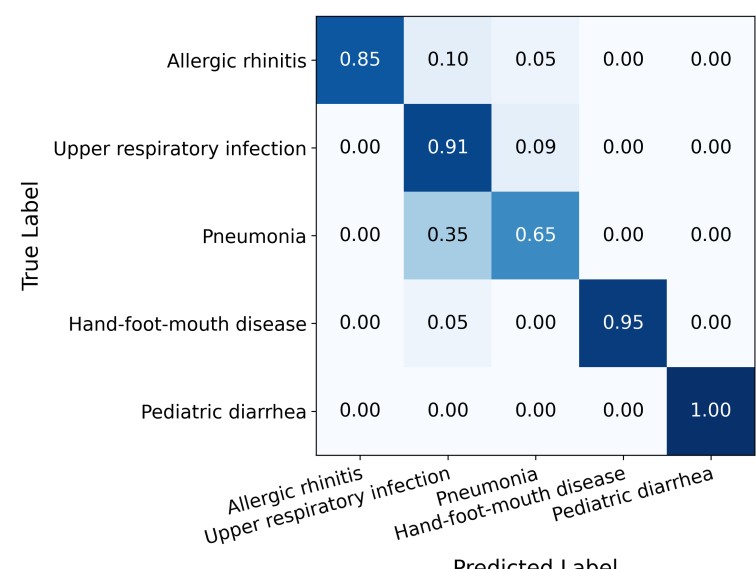

Figure 11: Confusion matrix for diagnostic performance on the Dxy dataset

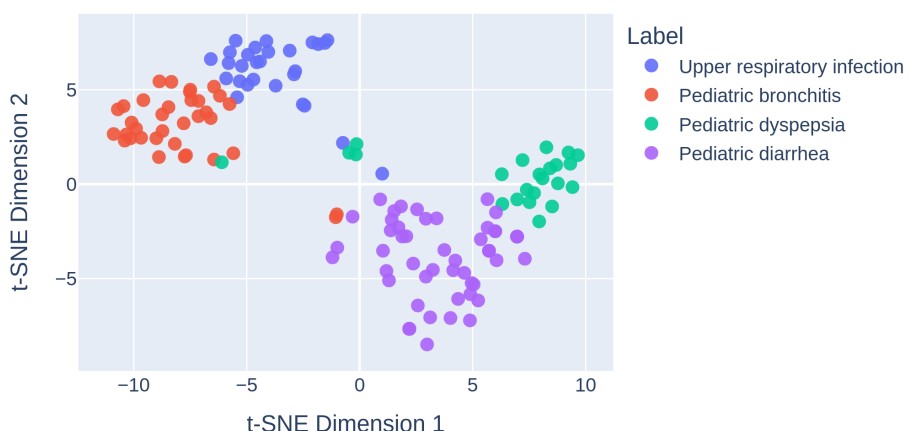

Figure 12: t-SNE visualization of the learned embedding representations on the MZ-4 Dataset

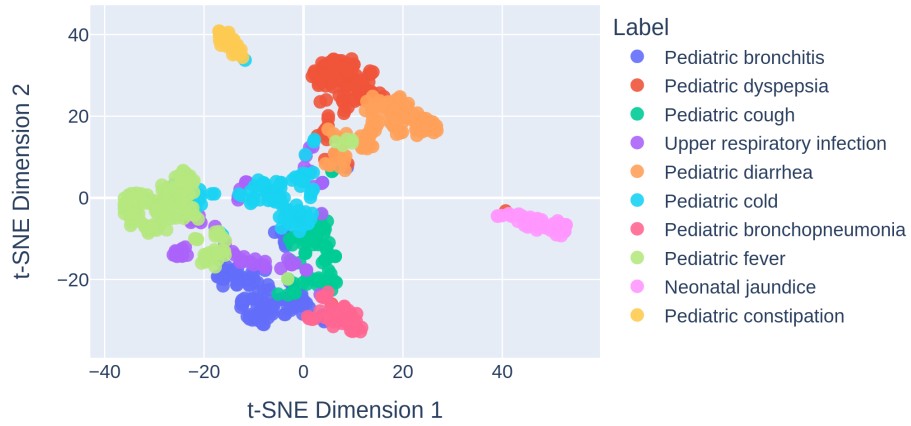

Figure 13: t-SNE visualization of the learned embedding representations on the MZ-10 Dataset

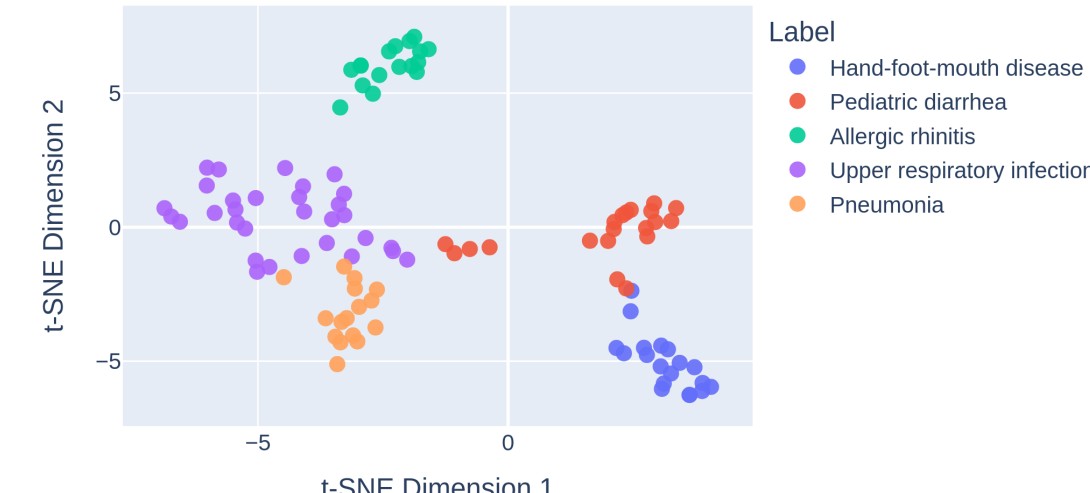

Figure 14: t-SNE visualization of the learned embedding representations on the Dxy Dataset

### A.6   PROMPT TEMPLATE

Figure 15 illustrates our prompt template designed to guide a large language model (LLM) in extracting medically relevant information from dialogue data. The model is directed to function as an advanced medical information extraction assistant, tasked with identifying symptom-disease pairs and representing them as structured semantic triples in the format: [Symptom, indicates, Disease], where "indicates" denotes the relationship between Symptom and Disease.

For its objective and function, the prompt transforms medical data into a set of triplets, enabling downstream applications such as KG construction and automated diagnosis. This structured format enhances interpretability, consistency, and integration into graph-based machine learning models like SPKGDIAG.

To ensure the quality and clinical relevance of extracted data, the prompt enforces several constraints, including medical relevance, comprehensive symptom coverage, fixed disease labels, clinical validity, and broad scope. For the output format, only a list of triples is returned, excluding any additional commentary, explanations, or formatting artifacts.

In its usage context, the prompt facilitates consistent and high-quality extraction of symptom semantics, laying the foundation for constructing patient-centric knowledge graphs that enhance diagnostic reasoning.

**Prompt Template**

**# Role and Instruction**

You are an advanced medical information extraction assistant. Given a patient-doctor conversation that discusses symptoms, diagnostic reasoning, and/or medical conditions, extract all medically relevant symptom terms and represent them as structured semantic triples.

Each triple must follow this format:

**[Symptom, indicates, Disease]**

**# Extraction Guidelines**

- Focus exclusively on meaningful, medically relevant symptoms. Ignore vague or unrelated terms.
- Make the most of the information given: extract both **direct** and **implied** symptoms, even if paraphrased or reworded.
- The **disease must remain fixed and singular** — use it exactly as written in the input.
- Ensure that every element in the triple ([Symptom, indicates, Disease]) is **clear, conclusive, and clinically valid**.
- Extract comprehensively — capture both **breadth (variety)** and **depth (granularity)** of symptoms.
- Output only the list of triples — no explanation, commentary, or formatting outside the list**.**

**# Example:**

**prompt:**

P: I've been having **chest pain**, especially when walking fast or climbing stairs. It feels like pressure on my chest.

D: Do you also feel **shortness of breath** during activity?

P: Yes

D: Do you often experience **dizziness**?

P: Yes

D: Have you noticed **swelling in your legs or ankles**, particularly in the evening?

P: Yes

D: Based on your symptoms, you may be dealing with a **circulatory system disease**.

**updates:** [[chest pain, indicates, circulatory system disease],
      [shortness of breath, indicates, circulatory system disease],
      [dizziness, indicates, circulatory system disease],
      [leg swelling, indicates, circulatory system disease]]

Now extract triples from the following input:

*Prompt: {dialogue_content}*

*Updates:*

Figure 15: Prompt template for symptom extraction

