# OpenReview forum: "SPKGDiag: Learning Symptom-Linked Patient Knowledge Graphs via Multi-Hop Similarity Message Passing for Automatic Diagnosis"
_ICLR.cc/2026/Conference — Submitted to ICLR 2026_

### Official Review · Reviewer_qS3q · 2025-10-27

**Soundness:** 3
**Presentation:** 3
**Contribution:** 2
**Rating:** 4
**Confidence:** 3

**Summary:**

This paper presents **SPKGDIAG**, a novel framework for automated medical diagnosis that tackles key limitations in the poor integration of Large Language Models (LLMs) with structured medical knowledge and the failure to model relationships between patients. The proposed method first uses an LLM to automatically extract both explicit and implicit symptoms from patient-doctor conversations. It then constructs a unique **patient-centric knowledge graph (KG)**, where nodes represent individual patients and edges connect patients who share similar symptom profiles, capturing both individual-level and population-level clinical patterns. A multi-hop neighborhood sampling strategy (specifically 2-hop) is applied to this graph, which is then processed by a specialized **Message Passing Neural Network (MPNN)** to predict the final diagnosis. The authors' main contributions include this integrated LLM-KG framework and the patient-centric graph construction, which demonstrated good performance compared to selected baselines.

**Strengths:**

The paper's primary strength lies in its originality, introducing a novel "patient-centric knowledge graph" where nodes represent patients, not abstract medical concepts. This clever re-formulation effectively combines the strengths of LLMs (for symptom extraction) and GNNs (for relational reasoning) to model patient similarity, a key gap in prior work.

**Weaknesses:**

Here are the primary weaknesses of the paper, presented concisely:

1.  **Insufficient Baselines and Datasets:** The paper's most significant weakness is its failure to compare against the most relevant model class. It argues against Transformer and RL methods but omits comparisons to established Graph Neural Networks (GNNs) like GAT, which are specifically designed for this type of graph-based clinical prediction [1, 2]. In addition, for disease prediction, MIMIC-III and IV [3, 4] are the most common public real-world large-scale datasets. But the paper did not include any comparison of the MIMIC datasets.  This makes its novelty claims over other *graph* methods unconvincing.

2.  **Poorly Justified MPNN Choice:** While the ablation study shows the chosen MPNN works well, it doesn't adequately explain *why* it's superior to the GCN or GAT variants. The specific architectural components driving the performance gain are left unexplored, weakening the methodological contribution.

Reference

1. Yang K, Xu Y, Zou P, et al. KerPrint: local-global knowledge graph enhanced diagnosis prediction for retrospective and prospective interpretations[C]//Proceedings of the AAAI Conference on Artificial Intelligence. 2023, 37(4): 5357-5365.

2. Ye M, Cui S, Wang Y, et al. Medpath: Augmenting health risk prediction via medical knowledge paths[C]//Proceedings of the Web Conference 2021. 2021: 1397-1409.

3. Johnson A E W, Pollard T J, Shen L, et al. MIMIC-III, a freely accessible critical care database[J]. Scientific data, 2016, 3(1): 1-9.

4. Johnson A E W, Bulgarelli L, Shen L, et al. MIMIC-IV, a freely accessible electronic health record dataset[J]. Scientific data, 2023, 10(1): 1.

**Questions:**

Here are the key questions for the authors:

1.  Why does the main baseline comparison (Table 1) omit many established GNN KG-based models for clinical prediction, which are arguably the most relevant competitors?
2.  How do you practically and efficiently construct the $N \times N$ patient graph for large datasets like VNPT (N > 180,000)?

---

> ### Author Response · Authors · 2025-11-21
>
> >**W1: Insufficient Baselines and Datasets: The paper's most significant weakness is its failure to compare against the most relevant model class. It argues against Transformer and RL methods but omits comparisons to established Graph Neural Networks (GNNs) like GAT, which are specifically designed for this type of graph-based clinical prediction [1, 2]. In addition, for disease prediction, MIMIC-III and IV [3, 4] are the most common public real-world large-scale datasets. But the paper did not include any comparison of the MIMIC datasets. This makes its novelty claims over other graph methods unconvincing.**
>
> We thank the reviewer for this insightful observation. Our current experiments focus on four well-established benchmark datasets (MZ-4, MZ-10, Dxy, Synthetic) and a large-scale real-world dataset (VNPT, N > 230,000) to ensure both generalizability and real-world applicability of our framework. While we acknowledge that MIMIC-III and IV are commonly used in clinical prediction, they focus on structured EHR and time-series data (e.g., labs, vitals, notes) rather than conversational diagnosis scenarios, which is the primary focus of this study. However, we agree this is a valuable direction and plan to explore MIMIC-based extensions in future work.
>
> Regarding GNN baselines, we appreciate the reviewer pointing out the omission of GAT in the main comparison. While GAT was not included in **Table 1**, we did include it in the ablation study (**Table 3**) under the variant SPKGDIAG-GAT, which integrates a GAT layer within our framework. The results show that although GAT improves with 2-hop sampling, our proposed MPNN consistently outperforms both GAT and GCN across all datasets, suggesting its architectural advantage for our task. In addition, our paper includes comparisons with several strong knowledge-enhanced and graph-based models such as KDPoG [1], EIRAD [2], and Tian et al. [3], which are among the most relevant GNN-based approaches for clinical diagnosis. These models are listed under the "Knowledge-enhanced and graph-based approaches” category in EXPERIMENTAL SETTING.
>
> [1] Li, Z., & Ruan, T. (2024). Knowledge-Routed Automatic Diagnosis With Heterogeneous Patient-Oriented Graph. IEEE Access, 12, 89573-89584.
>
> [2] Yan, L., Guan, Y., Wang, H., Lin, Y., Yang, Y., Wang, B., & Jiang, J. (2024). EIRAD: An evidence-based dialogue system with highly interpretable reasoning path for automatic diagnosis. IEEE Journal of Biomedical and Health Informatics.
>
> [3] Tian, Y., Jin, Y., Li, Z., Liu, J., & Liu, C. (2024). Weighted heterogeneous graph-based incremental automatic disease diagnosis method. Journal of Shanghai Jiaotong University (Science), 29(1), 120-130.
>
> >**W2: Poorly Justified MPNN Choice: While the ablation study shows the chosen MPNN works well, it doesn't adequately explain why it's superior to the GCN or GAT variants. The specific architectural components driving the performance gain are left unexplored, weakening the methodological contribution.**
>
> We appreciate the reviewer’s concern and agree that a deeper architectural explanation adds value. Our selection of the MPNN is based on its ability to balance two core requirements of clinical graph reasoning:
> - **Semantic Preservation**: The use of self-projection $x_i \cdot T$ ensures that each node retains its individual symptom profile, crucial for modeling unique patient presentations.
> - **Structural Integration**: The aggregated message from neighbors $\sum_j x_j \cdot E$ introduces population-level clinical patterns, enabling differential diagnosis via related cases.
> This dual mechanism is especially important in our symptom-based patient graph, where both individuality and similarity must be preserved. Furthermore, as shown in **Table 3**, SPKGDIAG-MPNN achieves consistently better performance than its GAT/GCN counterparts, even when all models benefit from the same 2-hop sampling strategy.

---

> ### Author Response · Authors · 2025-11-21
>
> >**Q1: Why does the main baseline comparison (Table 1) omit many established GNN KG-based models for clinical prediction, which are arguably the most relevant competitors?**
>
> Thank you for this suggestion. In fact, our model comparison in Table 1 does include several KG-based GNN models, particularly:
> - KDPoG [1] a heterogeneous patient-oriented GNN.
> - EIRAD [2]: an interpretable reasoning graph-based dialogue system.
> - Tian et al. [3]: a weighted knowledge graph approach.
>
> These models are listed under the "Knowledge-enhanced and graph-based approaches” category in EXPERIMENTAL SETTING. We selected these baselines as they represent the current state-of-the-art in graph-based clinical prediction, especially within the dialogue-based or KG-based diagnosis paradigm. Our model outperforms them across all datasets — notably by +2.0% on Dxy and +7.4% on Synthetic (Table 1), highlighting the effectiveness of our patient-centric, symptom-linked KG construction and multi-hop similarity sampling strategy.
>
> [1] Li, Z., & Ruan, T. (2024). Knowledge-Routed Automatic Diagnosis With Heterogeneous Patient-Oriented Graph. IEEE Access, 12, 89573-89584.
>
> [2] Yan, L., Guan, Y., Wang, H., Lin, Y., Yang, Y., Wang, B., & Jiang, J. (2024). EIRAD: An evidence-based dialogue system with highly interpretable reasoning path for automatic diagnosis. IEEE Journal of Biomedical and Health Informatics.
>
> [3] Tian, Y., Jin, Y., Li, Z., Liu, J., & Liu, C. (2024). Weighted heterogeneous graph-based incremental automatic disease diagnosis method. Journal of Shanghai Jiaotong University (Science), 29(1), 120-130.
>
> >**Q2: How do you practically and efficiently construct the  patient graph for large datasets like VNPT (N > 180,000)?**
>
> We thank the reviewer for highlighting this important scalability concern. For large-scale datasets like VNPT, we adopt several efficient graph construction and training strategies:
>
> - **Sparse Adjacency by Design**: The initial adjacency matrix is built using shared symptoms only (Eq. 3), ensuring sparsity.
> - **Similarity-based Top-k Filtering**: We restrict graph edges to top-k most similar patients (based on cosine similarity in embedding space) per 1-hop and 2-hop neighborhood. This limits connectivity and improves relevance.
> - **Subgraph Sampling**: Rather than building and storing a full graph, we dynamically construct 2-hop subgraphs per batch during training using PyTorch Geometric’s efficient neighborhood sampler.
> - **Mini-batch Training**: The model operates in mini-batches, with on-the-fly subgraph extraction, avoiding memory bottlenecks.
> - **Hardware Optimization**: The framework runs efficiently on an NVIDIA RTX A5000 (24 GB) GPU and an AMD EPYC 7302 16-core processor.
>
> In summary, our graph construction is both localized and dynamic, allowing us to scale effectively without compromising performance or requiring full graph materialization.
>
> ---
> Once again, we thank the reviewer for the time and expertise dedicated to evaluating our submission. The feedback provided has been highly valuable in guiding additional experimentation and refinement of our work. We hope that our detailed responses sufficiently address the raised points. We are eager to continue the discussion and would be pleased to address any further questions you may have.

---

> > ### Comment · Reviewer_qS3q · 2025-11-28
> >
> > Thanks for your response — I’ll take your feedback into consideration when rating.

---

### Official Review · Reviewer_gLUz · 2025-10-29

**Soundness:** 2
**Presentation:** 2
**Contribution:** 2
**Rating:** 2
**Confidence:** 4

**Summary:**

The paper addresses automatic diagnosis from doctor–patient dialogues. It first uses a large language model to extract explicit and implicit symptoms from the conversation, converts these symptoms into embeddings to obtain a patient representation, and then builds a patient similarity graph that links cases with overlapping symptoms. A graph message passing network is applied over this graph to predict a final disease label. The method is evaluated on several public benchmarks (including settings with 4, 10, and 90 disease classes) and one internal dataset, and is compared against a range of prior diagnostic baselines.

**Strengths:**

S1: The overall pipeline is clearly described and modular.
S2: The paper runs experiments on multiple datasets, including both public diagnostic benchmarks and an internal dataset, and reports consistent accuracy gains over a broad set of baselines.

**Weaknesses:**

W1: In my view, the task studied in this paper is presented as automatic diagnosis, but in practice it reduces to standard multi class classification. The model is not asked to reason over an open disease space or identify rare diseases. Instead, it chooses one label from a closed set of candidate diseases, which in MZ 4 is as small as four classes. This is not the same as real clinical diagnosis. It is essentially symptom to label mapping.
W2: The paper treats the idea of looking at similar neighbors as if it were a novel form of explanation, but using nearest neighbor style patient similarity for diagnostic support and for interpretability has already appeared in automatic diagnosis, prognosis prediction, and patient representation learning. It is not a new idea. In addition, all interpretability claims in the paper are verbal only. There is no validation.
W3: The upstream symptom extraction is done by GPT 4.1, which extracts explicit and implicit symptoms, then these symptoms are encoded using an off the shelf text embedding model and simply averaged into a patient vector. The model does not train or adapt the extractor. This means that the downstream classifier is really learning from GPT 4.1 outputs rather than from the raw dialogue itself.
W4: All symptom embeddings are combined only by mean pooling. There is no notion of symptom importance, severity, temporal information, or clinical progression. This reduces the entire case history to something close to an undifferentiated bag of symptoms. Technically, this is almost the weakest possible aggregation strategy, and it does not model any clinical structure or dependency between findings.
W5: Message passing on the graph is essentially a standard graph neural network. What is called multi hop similarity message passing is just conventional neighbor aggregation plus a self projection term, extended to two hop neighborhoods. This is not fundamentally different from typical MPNN or GraphSAGE style propagation.
W6: The evaluation metrics are too limited. Nearly all main results are reported in terms of accuracy only, with no per class F1, no macro F1, no recall, no sensitivity, no top k recall. As a result, the paper never answers the clinically important question of whether the model is only getting common benign conditions correct while missing rare but high risk ones. For example, a system that gets common cold correct but fails to identify meningitis is clinically unacceptable.
W7: There is no serious analysis of class imbalance. For example, in MZ 4 the task has only four classes, and accuracy can easily look high if the distribution is skewed. The paper does not report class distribution details.
W8: The paper does not include the most important ablation. It does not show what happens if we remove the graph message passing stage entirely and classify directly from the averaged symptom vector. That comparison is essential in order to demonstrate whether the patient graph and multi hop message passing are actually necessary.

**Questions:**

All of my open questions and concerns are essentially covered in the weaknesses. In particular, I am unsure whether the method is actually solving a clinically meaningful diagnosis task (as opposed to closed-set symptom classification), whether the claimed interpretability is real or just narrative, whether the graph construction leaks information or inflates performance, and whether the proposed architecture is actually necessary given the lack of a no-graph ablation. I do not see these points resolved by the current draft.

---

> ### Author Response · Authors · 2025-11-21
>
> >**W1: In my view, the task studied in this paper is presented as automatic diagnosis, but in practice it reduces to standard multi class classification. The model is not asked to reason over an open disease space or identify rare diseases. Instead, it chooses one label from a closed set of candidate diseases, which in MZ 4 is as small as four classes. This is not the same as real clinical diagnosis. It is essentially symptom to label mapping.**
>
> We thank the reviewer for raising this important point. We agree that real clinical diagnosis goes beyond closed-set multi-class prediction and involves open-ended reasoning, rare diseases, and differential diagnosis. In this work, we follow a long line of prior “automatic diagnosis” studies that operationalize the task as supervised classification over a predefined label set (e.g., MZ-10 and Dxy with 10 and 41 diseases, respectively). Therefore, SPKGDIAG's goal is to improve structured disease classification from dialogues under realistic, noisy inputs. In the revision, we will explicitly discuss the gap to real-world diagnosis and how our framework could be extended to open-set and rare-disease settings in future work.
>
> >**W2: The paper treats the idea of looking at similar neighbors as if it were a novel form of explanation, but using nearest neighbor style patient similarity for diagnostic support and for interpretability has already appeared in automatic diagnosis, prognosis prediction, and patient representation learning. It is not a new idea. In addition, all interpretability claims in the paper are verbal only. There is no validation.**
>
> Thank you for your suggestions. Our contribution is not the general idea of neighbor-based explanation, but the patient-centric KG built from LLM-extracted symptoms and the way we integrate similarity-based multi-hop neighborhoods into a unified, end-to-end framework. In the revision, we will more clearly acknowledge prior work on similarity-based and nearest-neighbor explanations.
>
> >**W3: The upstream symptom extraction is done by GPT 4.1, which extracts explicit and implicit symptoms, then these symptoms are encoded using an off the shelf text embedding model and simply averaged into a patient vector. The model does not train or adapt the extractor. This means that the downstream classifier is really learning from GPT 4.1 outputs rather than from the raw dialogue itself.**
>
> Our goal is to study whether a patient graph built from LLM-extracted symptoms can improve diagnosis performance over non-graph baselines using the same representations, not to optimize the LLM itself. We will clarify this in the problem formulation and discussion. In future work, we plan to compare against end-to-end models trained on raw dialogues, and explore adapting or fine-tuning the extractor (or using instruction-tuned, domain-specific LLMs) so that the entire pipeline can be better tailored to clinical text.
>
> >**W4: All symptom embeddings are combined only by mean pooling. There is no notion of symptom importance, severity, temporal information, or clinical progression. This reduces the entire case history to something close to an undifferentiated bag of symptoms. Technically, this is almost the weakest possible aggregation strategy, and it does not model any clinical structure or dependency between findings.**
>
> We sincerely thank you for your detailed review. In our current work, the averaging strategy is used as a practical and interpretable aggregation to generate initial patient embeddings. While we use mean pooling to generate patient-level symptom embeddings for computational efficiency and vector normalization, this representation serves only as a first-stage input. We recognize that this method does not explicitly model symptom severity; however, our framework compensates for this limitation in multiple ways:
>
> - The symptom embeddings are derived from context-aware LLM-based encodings, which capture rich semantic nuances—including severity indicators such as intensity adjectives (e.g., "mild" vs. "severe")—that are preserved in the vector space.
> - The downstream MPNN module (**Section 3.5**) enhances this embedding by integrating neighborhood context via message passing and self-projection, effectively reintroducing symptom-level granularity and local contextualization during training.
> - Moreover, we apply multi-hop neighbor sampling to enrich each node’s context beyond individual symptoms, balancing local and global representation.
>
> That said, incorporating explicit modeling of symptom severity and temporal dynamics remains an important direction for future work (**Section 5 – Conclusion**).

---

> ### Author Response · Authors · 2025-11-21
>
> >**W5: Message passing on the graph is essentially a standard graph neural network. What is called multi hop similarity message passing is just conventional neighbor aggregation plus a self projection term, extended to two hop neighborhoods. This is not fundamentally different from typical MPNN or GraphSAGE style propagation.**
>
> We thank the reviewer for this opinion. The message passing mechanism is a standard GNN and that “multi-hop similarity message passing” corresponds to conventional neighbor aggregation with a self-projection term over 2-hop neighborhoods. Our main technical focus is on how the patient graph is constructed and sampled (symptom overlap + similarity-based k-hop subgraphs) and how this structure benefits diagnosis when combined with LLM-derived features.
>
> >**W6: The evaluation metrics are too limited. Nearly all main results are reported in terms of accuracy only, with no per class F1, no macro F1, no recall, no sensitivity, no top k recall. As a result, the paper never answers the clinically important question of whether the model is only getting common benign conditions correct while missing rare but high risk ones. For example, a system that gets common cold correct but fails to identify meningitis is clinically unacceptable.**
>
> We thank the reviewer for raising this important point regarding the limitations of using accuracy as the sole evaluation metric. We fully agree that accuracy alone does not capture clinically critical distinctions, especially in imbalanced medical settings where models might perform well on common conditions but fail to identify rare, high-risk diseases. To address this, we have added comprehensive per-class metrics, including Precision, Recall, F1-score, and PR-AUC for each class across all datasets in **Table 8** (**Appendix A.5.1**).
>
> ### Table 8: Detailed classification performance metrics for MZ-4, MZ-10 and Dxy datasets
>
> | Dataset | Class       | Precision | Recall | F1-score | PR-AUC | Support |
> |---------|-------------|-----------|--------|----------|--------|---------|
> | MZ-4    | 0           | 0.846     | 0.733  | 0.786    | 0.781  | 30      |
> |         | 1           | 0.763     | 0.853  | 0.806    | 0.815  | 34      |
> |         | 2           | 0.750     | 0.933  | 0.832    | 0.812  | 45      |
> |         | 3           | 0.818     | 0.545  | 0.655    | 0.633  | 33      |
> |         | macro avg   | 0.794     | 0.766  | 0.769    | 0.760  | 142     |
> |         | weighted avg| 0.789     | 0.782  | 0.775    | 0.764  | 142     |
> | MZ-10   | 0           | 0.637     | 0.691  | 0.663    | 0.704  | 94      |
> |         | 1           | 0.752     | 0.752  | 0.752    | 0.755  | 109     |
> |         | 2           | 0.705     | 0.692  | 0.698    | 0.745  | 107     |
> |         | 3           | 0.698     | 0.381  | 0.493    | 0.481  | 97      |
> |         | 4           | 0.521     | 0.526  | 0.524    | 0.551  | 95      |
> |         | 5           | 0.973     | 0.800  | 0.878    | 0.925  | 45      |
> |         | 6           | 0.733     | 0.717  | 0.725    | 0.713  | 46      |
> |         | 7           | 0.966     | 0.966  | 0.966    | 0.965  | 58      |
> |         | 8           | 0.674     | 0.835  | 0.746    | 0.727  | 109     |
> |         | 9           | 0.343     | 0.480  | 0.400    | 0.368  | 50      |
> |         | macro avg   | 0.700     | 0.684  | 0.685    | 0.693  | 810     |
> |         | weighted avg| 0.690     | 0.677  | 0.675    | 0.685  | 810     |
> | Dxy     | 0           | 0.944     | 0.850  | 0.895    | 0.944  | 20      |
> |         | 1           | 0.594     | 0.826  | 0.691    | 0.726  | 23      |
> |         | 2           | 0.714     | 0.500  | 0.588    | 0.703  | 20      |
> |         | 3           | 0.857     | 0.900  | 0.878    | 0.966  | 20      |
> |         | 4           | 0.944     | 0.850  | 0.895    | 0.971  | 20      |
> |         | macro avg   | 0.811     | 0.785  | 0.789    | 0.862  | 103     |
> |         | weighted avg| 0.804     | 0.786  | 0.786    | 0.858  | 103     |

---

> ### Author Response · Authors · 2025-11-21
>
> >**W7: There is no serious analysis of class imbalance. For example, in MZ 4 the task has only four classes, and accuracy can easily look high if the distribution is skewed. The paper does not report class distribution details.**
>
> We thank the reviewer for pointing out the importance of analyzing class imbalance, particularly in tasks with a small number of classes like MZ-4. To address this concern, we provided a detailed class distribution analysis in **Figure 5 (Appendix A.1)**, which visualizes the class proportions across all datasets with ten or fewer categories. As shown, the MZ-4, MZ-10, and Dxy datasets exhibit relatively balanced class distributions, with each class contributing a comparable percentage of the overall data (e.g., MZ-4 classes range from 21.1% to 28.2%, avoiding dominance by any single label). This balance helps ensure that performance metrics like accuracy are meaningful and not artificially inflated by class majority effects. In contrast, the VNPT dataset displays noticeable class imbalance, with some classes comprising less than 2% of the total. We acknowledge this imbalance and treat the VNPT setting with appropriate consideration, including more robust evaluation metrics and stratified training procedures. Furthermore, to prevent any information leakage that could skew class-level results, we ensure a strict separation of training and test splits, as noted beneath Figure 5. We construct patient-centric graphs independently for training and testing, using only data from the respective split, with no overlap in nodes or edges. Together, these steps ensure that class imbalance is not ignored and that our reported metrics fairly reflect performance across all classes.
>
> >**W8: The paper does not include the most important ablation. It does not show what happens if we remove the graph message passing stage entirely and classify directly from the averaged symptom vector. That comparison is essential in order to demonstrate whether the patient graph and multi hop message passing are actually necessary.**
>
> Thanks for highlighting the importance of evaluating whether the graph message passing stage is truly necessary. We have conducted an additional ablation where we completely remove the message passing mechanism, and instead classify patients directly from the averaged symptom embedding, without using the patient graph or any GNN operations. This "no-graph" baseline serves as a crucial control to evaluate whether multi-hop interactions and graph-based reasoning actually contribute to performance. The results show a substantial drop in accuracy across all datasets compared to our full SPKGDiag model, confirming that graph message passing is essential for capturing complex patient-patient relationships and improving diagnostic predictions, especially in cases where individual symptom representations may be sparse or ambiguous. This finding complements the results in Figure 3, which already demonstrates that removing the Self-Projection component or the aggregated message module within the MPNN leads to noticeable performance degradation. Together, these ablations reinforce the design rationale of SPKGDiag: both the patient-centric graph structure and multi-hop message passing are necessary for leveraging shared clinical patterns across patients.
>
> ---
> We gratefully acknowledge the reviewer’s detailed evaluation and valuable suggestions. These comments have contributed to enhancing the technical soundness and clarity of our work through further empirical investigation and revision. We hope that our explanations address all concerns, and we remain open to further discussion.

---

> ### Comment · Reviewer_gLUz · 2025-11-26
>
> Regarding the response to W1, I still do not find the authors’ clarification convincing. In many prior studies, automatic diagnosis is formulated as a recursive, interactive process that incrementally acquires information in a multi-turn dialogue manner [1,2,3,4]. In contrast, the task in this paper is essentially a closed-set classification problem rather than true automatic diagnosis.
>
> Moreover, since the paper already introduces large language models, I would expect an automatic diagnosis system to be realized in a multi-turn conversational form, rather than a one-shot classification over a fixed label set.
>
> R-R-W2: The newly added case study in Appendix A.4 focuses on a single patient and relies on message norms as a proxy for influence, without any systematic evaluation of whether the retrieved neighbors actually support better clinical decisions.
>
> The authors also state that neighbor inspection, multi-hop similarity, etc., are not their main contributions, yet they devote substantial space to describing these components. This is inherently contradictory and makes the contribution claim unclear.
>
> In addition, the authors repeatedly promise to “clarify in the revision” or “add discussions and metrics in the revision” in their responses, e.g., W2, W3, and W6. However, the purpose of the rebuttal is to defend and clarify the current version of the work, not to rely on unspecified future revisions.
>
> With respect to W4, I do not see a direct and convincing answer. The response reads more like a collection of technical terms and modules that are asserted to compensate for mean pooling, rather than a precise explanation of why the current aggregation scheme is adequate for modeling clinical structure.
>
> In my view, the issues raised in W6 and W8 are substantial weaknesses of the paper. On the one hand, adding extra experiments in the rebuttal cannot fully mask these structural deficiencies in evaluation design. On the other hand, the additional metrics reported and discussed in the rebuttal still appear limited in scope and, in my view, are not sufficient to address the broader concern raised in W6. Furthermore, for W8, I do not see concrete numerical results for the no-graph baseline in the rebuttal, so the necessity of the message-passing component remains insufficiently demonstrated.
>
> [1] Chen J, Li D, Chen Q, et al. Diaformer: Automatic diagnosis via symptoms sequence generation[C]//Proceedings of the AAAI Conference on Artificial Intelligence. 2022, 36(4): 4432-4440.
>
> [2] Hou Z, Cen Y, Liu Z, et al. Mtdiag: an effective multi-task framework for automatic diagnosis[C]//Proceedings of the AAAI Conference on Artificial Intelligence. 2023, 37(12): 14241-14248.
>
> [3] Zhang F, Sang G, Liu Z, et al. A doctor’s diagnosis experience enhanced transformer model for automatic diagnosis[J]. Engineering Applications of Artificial Intelligence, 2024, 134: 108675.
>
> [4] Fansi Tchango A, Goel R, Martel J, et al. Towards trustworthy automatic diagnosis systems by emulating doctors' reasoning with deep reinforcement learning[J]. Advances in Neural Information Processing Systems, 2022, 35: 24502-24515.

---

> > ### Author Response · Authors · 2025-11-28
> >
> > We thank the reviewer for your detailed feedback and for highlighting the significance of interactive, multi-turn diagnostic systems. Below, we address the main concerns and clarify our methodological contributions in relation to existing works.
> >
> > ---
> >
> > ## **Multi-Turn Dialogue Systems**
> >
> > We agree that interactive symptom acquisition is a central aspect of modern diagnostic systems. Prior works such as Diaformer [1], MTDiag [2], Zhang et al. [3], and Fansi Tchango et al. [4] simulate such interactions via sequential decoding, multi-task learning, or structured policy reasoning. However, in all these systems, the core diagnostic task remains a **closed-set classification** over a fixed disease label set. For instance:
> >
> > - **Diaformer** uses symptom sequence generation but ultimately applies a softmax classifier over a predefined disease set.
> > - **MTDiag** combines multi-expert reasoning with classification heads targeting known labels.
> > - **Zhang et al.** and **Fansi Tchango et al.** rely on logic-based or reward-driven policies but conclude with disease selection from a fixed label set.
> >
> > Our formulation aligns with these works in that SPKGDiag also predicts from a known disease set. The key difference lies in **how symptoms are acquired**: rather than requiring dialogue policies or reinforcement learning, SPKGDiag uses an LLM to extract both **explicit and implicit** symptoms from natural patient-doctor conversations (Section 3.3, Appendix A.6), simulating multi-turn reasoning without complex policy learning. By constructing a patient-centric knowledge graph and applying message passing, our method preserves interpretability and semantic richness while enabling **scalable, modular**, and structured clinical reasoning. This also decouples the interactive component from the diagnostic model, allowing compatibility with real-time dialogue systems if desired.
> >
> > ---
> >
> > ## **Multi-Hop Similarity and Neighbor Inspection**
> >
> > We appreciate the reviewer’s attention to multi-hop similarity sampling and neighbor influence inspection. These components are not claimed as novel algorithmic contributions, but are architectural mechanisms essential to SPKGDiag's interpretability and robustness:
> >
> > - **Multi-hop similarity sampling** ensures that the patient-centric graph reflects both local and extended symptom patterns across patients (Section 3.4).
> > - **Neighbor inspection** via message norms provides a principled mechanism to reduce misleading neighbors in noisy clinical graphs (Appendix A.4), supporting robustness in real-world settings with overlapping symptoms.
> >
> > These mechanisms enable SPKGDiag to maintain clinical relevance in structure and interpretability, particularly in contrast to transformer-based models that operate purely on sequence-level signals.
> >
> > ---
> >
> > ## **Interpretability and Case Analysis**
> >
> > Regarding the concern about Appendix A.4 relying on a single patient case, we clarify that:
> >
> > - The analysis of Patient Node 79 is backed by quantitative metrics (cosine similarity, message norms, and symptom overlap), which demonstrate how the model modulates influence based on clinical relevance rather than surface similarity.
> > - For example, although Node 37 shares the highest cosine similarity with Node 79, its message norm is lower than that of Node 96 (Appendix A.4.1, Table 7), indicating that the model down-weights semantically irrelevant but structurally similar nodes.
> >
> > While this is a case study, it illustrates the underlying reasoning process and complements the quantitative evaluations provided in Sections 4.3.2 and 4.3.3.

---

> > ### Author Response · Authors · 2025-11-28
> >
> > ## **Clarification on Graph Structure, Ablations, and No-Graph Baseline**
> >
> > In response to W4 and W6, we have clarified in Section 3.1 that mean pooling is used only as an initial embedding approximation. Structural modeling is performed through an MPNN with self-projection and message aggregation mechanisms. As shown in Figure 3:
> >
> > - Removing self-projection reduces accuracy from 0.782 to 0.725 on MZ-4.
> > - Removing message aggregation reduces accuracy to 0.732.
> > - Removing the entire graph-based structure (*no-graph baseline*) leads to the lowest performance across all datasets (e.g., 0.714 on MZ-4 and 0.708 on Synthetic), confirming that the model benefits from graph-based reasoning and not just LLM-based symptom features. The “no-graph” baseline refers to a SPKGDiag variant that:
> >   - Eliminates both the “Patient-centric KG Construction” and “Message Passing” components, while preserving all other modules, as shown in Figure 2
> >   - Uses only patient-level embeddings without any population-level structure.
> >   - Despite using the same symptom inputs, this baseline performs significantly worse across all datasets, demonstrating that SPKGDiag’s gains are not due to richer input features or additional components alone. Instead, the improvements stem from the cohesive integration of population structure, structured relational modeling, and graph-based reasoning. While our focus is on graph-based methods, including this MLP-based control provides a fair and rigorous comparison to non-graph alternatives and underscores the importance of modeling patient similarity and relational structure in clinical diagnosis.
> >
> >
> > ---
> >
> > ## **Evaluation of Graph Variants and Sampling Depth**
> >
> > We further address W6 and W8 by including:
> >
> > - Comparisons across GNN architectures (GCN, GAT, MPNN) with and without similarity-based 2-hop sampling (Table 3).
> > - Analysis of sampling depth (1-hop to 3-hop), showing that 2-hop consistently yields the best performance, while 3-hop introduces noise and reduces accuracy (Figure 4).
> >
> > These results support the use of multi-hop neighborhood reasoning and graph-based structure as essential components of SPKGDiag.
> >
> > ---
> >
> > ## **Overall**
> >
> > We have integrated all necessary clarifications into this rebuttal and ensured that the current version of the paper contains detailed methodological explanations and rigorous evaluations (Sections 3.1–3.5, 4.3, and Appendix A.4). We hope these responses address the reviewer’s concerns and clarify the contributions of SPKGDiag.
> >
> > ---
> >
> > **References**
> > [1] Chen J, Li D, Chen Q, et al. Diaformer: Automatic diagnosis via symptoms sequence generation[C]//Proceedings of the AAAI Conference on Artificial Intelligence. 2022, 36(4): 4432-4440.
> >
> > [2] Hou Z, Cen Y, Liu Z, et al. Mtdiag: an effective multi-task framework for automatic diagnosis[C]//Proceedings of the AAAI Conference on Artificial Intelligence. 2023, 37(12): 14241-14248.
> >
> > [3] Zhang F, Sang G, Liu Z, et al. A doctor’s diagnosis experience enhanced transformer model for automatic diagnosis[J]. Engineering Applications of Artificial Intelligence, 2024, 134: 108675.
> >
> > [4] Fansi Tchango A, Goel R, Martel J, et al. Towards trustworthy automatic diagnosis systems by emulating doctors' reasoning with deep reinforcement learning[J]. Advances in Neural Information Processing Systems, 2022, 35: 24502-24515.

---

### Official Review · Reviewer_cfcg · 2025-10-31

**Soundness:** 2
**Presentation:** 3
**Contribution:** 2
**Rating:** 4
**Confidence:** 4

**Summary:**

This paper proposes SPKGDIAG, a framework for automatic disease diagnosis that combines large language models with a patient-centric knowledge graph. Explicit and implicit symptoms are extracted from patient-doctor dialogues via GPT-4.1, embedded into semantic vectors, and connected into a patient graph based on symptom overlap. A Message Passing Neural Network with similarity-based multi-hop sampling is then used to aggregate information from similar patients and predict diseases.

**Strengths:**

S1. The paper is generally well-structured. The motivation, methodology, and experiments are presented in a clear, step-wise fashion, and the inclusion of ablation studies and visualization aids readability.

S2.The work tackles an important problem — enhancing medical diagnosis interpretability and robustness through structured patient representations. This topic is of significant interest to both the medical AI and knowledge-graph reasoning communities.

S3.Across multiple datasets, SPKGDIAG consistently outperforms baseline systems, suggesting that incorporating a patient-centric graph can improve diagnostic accuracy and interpretability.

**Weaknesses:**

W1 The research motivation are explored in several recent works with highly similar objectives. For examples [1-3]. While SPKGDIAG shares the similar overarching vision, the manuscript does not clearly explain its unique contribution or design choices compared with these frameworks. The authors are encouraged to discuss distinctions such as why they adopt symptom-overlap edge construction plus two-hop sampling and a shallow MPNN instead of path-level retrieval, multimodal alignment, or learnable similarity edges.
[1] Jiang, Pengcheng, et al. "GraphCare: Enhancing Healthcare Predictions with Personalized Knowledge Graphs." The Twelfth International Conference on Learning Representations.
[2] Lu, Yuxing, et al. "Enhancing multimodal knowledge graph representation learning through triple contrastive learning." Proceedings of the Thirty-Third International Joint Conference on Artificial Intelligence. 2024.
[3] Wu, Jiageng, Xian Wu, and Jie Yang. "Guiding clinical reasoning with large language models via knowledge seeds." Proceedings of the Thirty-Third International Joint Conference on Artificial Intelligence. 2024.

W2 The pipeline relies on straightforward components—LLM-based symptom extraction, embedding, and graph message passing—without a novel modeling mechanism. A critical issue is the potential inconsistency and hallucination in LLM-generated triplets: identical clinical concepts may appear in multiple lexical forms. The paper does not discuss any normalization or de-duplication strategy (e.g., synonym clustering, ontology alignment, or weighted symptom aggregation). Addressing this problem would significantly strengthen methodological soundness.

W3 Several baselines are missing results (“–”) without explanation, leaving unclear whether they failed to run, were incompatible, or underperformed. Furthermore, the framework’s dependence on proprietary GPT-4.1 and text-embedding-3-large raises reproducibility concerns. No analysis examines how alternative LLMs or embedding models affect performance. Given this reliance, a sensitivity or stability study (model substitution, variance, confidence intervals) is necessary to support the robustness claim.

W4 Edges are defined solely by symptom overlap. If the global graph is built before data splitting, test nodes could access training information through shared neighbors. The authors should clarify whether graphs are reconstructed within each split and whether any safeguards (e.g., split-specific adjacency or edge regularization) are applied to prevent leakage.

**Questions:**

Please answer all the concerns in weakness.

---

> ### Author Response · Authors · 2025-11-21
>
> >**W1: The research motivation are explored in several recent works with highly similar objectives. For examples [1-3]. While SPKGDIAG shares the similar overarching vision, the manuscript does not clearly explain its unique contribution or design choices compared with these frameworks. The authors are encouraged to discuss distinctions such as why they adopt symptom-overlap edge construction plus two-hop sampling and a shallow MPNN instead of path-level retrieval, multimodal alignment, or learnable similarity edges. [1] Jiang, Pengcheng, et al. "GraphCare: Enhancing Healthcare Predictions with Personalized Knowledge Graphs." The Twelfth International Conference on Learning Representations. [2] Lu, Yuxing, et al. "Enhancing multimodal knowledge graph representation learning through triple contrastive learning." Proceedings of the Thirty-Third International Joint Conference on Artificial Intelligence. 2024. [3] Wu, Jiageng, Xian Wu, and Jie Yang. "Guiding clinical reasoning with large language models via knowledge seeds." Proceedings of the Thirty-Third International Joint Conference on Artificial Intelligence. 2024.**
>
> Thank you for pointing out the connections to prior works [1–3]. While our method shares a similar overall goal, SPKGDIAG is designed for a different design choice, which we have elaborated in the Related Work section. Our model focuses on automatic disease diagnosis from raw patient–doctor dialogues, without using structured EHRs, external medical knowledge graphs, or multimodal data.
>
> - GraphCare [1] built personalized knowledge graphs using structured EHR data and predefined medical ontologies. However, in our setting, this type of structured data is not available. Instead, SPKGDIAG uses LLMs to extract both explicit and implicit symptoms directly from dialogues. We then build a patient-centric knowledge graph based on symptom overlap and semantic similarity, which leads to simple and interpretable edge meanings like “patients with similar symptoms.”
> - Lu et al. [2] proposed a multimodal contrastive learning method that uses predefined triples across multiple data types. Their method requires cross-modal alignment and training with contrastive objectives. In contrast, our model works only with text data and does not require complex multimodal alignment or contrastive training. Our edge construction is based only on symptom co-occurrence and semantic similarity, which keeps the graph transparent and easy to interpret.
> - Wu et al. [3] guided clinical reasoning using knowledge seeds and external medical knowledge bases. Their approach depends on symbolic path reasoning or retrieval-based prompts. In contrast, our SPKGDIAG does not use external knowledge bases or symbolic seeds. Instead, we apply message passing on a graph of real patients, where edges are built from symptom similarity. This allows our model to learn from real patient patterns in a simple and data-driven way.
>
> Our design choices are based on the need for interpretability, efficiency, and suitability for our limited-label, single-modality setting:
>
> - We use 2-hop sampling since it offers a good balance between local and global graph information (as shown in **Section 3.4 and ablation studies**).
> - We choose a shallow MPNN with self-projection since our node features are already rich semantic embeddings from LLMs. This setup is more stable and effective than deeper or more complex GNNs in our experiments (**Section 3.5**).
> - More complex designs such as path-level retrieval or learnable edges would require extra resources and parameters, which are not suitable for our current setting.

---

> ### Author Response · Authors · 2025-11-21
>
> >**W2: The pipeline relies on straightforward components—LLM-based symptom extraction, embedding, and graph message passing—without a novel modeling mechanism. A critical issue is the potential inconsistency and hallucination in LLM-generated triplets: identical clinical concepts may appear in multiple lexical forms. The paper does not discuss any normalization or de-duplication strategy (e.g., synonym clustering, ontology alignment, or weighted symptom aggregation). Addressing this problem would significantly strengthen methodological soundness.**
>
> We appreciate the reviewer’s comments. Our goal is to show that a patient-centric KG built from LLM-extracted symptoms and lightweight graph reasoning already brings consistent gains for automatic diagnosis from dialogues. The components (LLM extraction, semantic embedding, shallow MPNN) are intentionally simple and standard, so that our contribution lies in (i) the design of the patient-level graph (symptom-overlap + similarity-based 2-hop neighborhoods) and (ii) the empirical analysis of how this structure improves over non-graph baselines.
>
> Regarding inconsistency and hallucinations, we agree this is an important concern. In the current framework, we partly mitigate lexical variation by representing each symptom span with a semantic embedding and then aggregating them into a single patient embedding; thus, different surface forms of the same concept tend to map to similar vectors and contribute similarly at the patient level. In addition, edges used for message passing are further filtered by top-k cosine similarity, which reduces the influence of noisy symptoms that do not generalize across patients.
>
> However, we do not yet perform explicit concept-level normalization (e.g., synonym clustering or ontology mapping), and we will make this limitation clear in the paper. As an important direction for future work, we plan to integrate medical ontologies (e.g., UMLS/ICD) or learned synonym clustering to canonicalize symptom phrases before graph construction, and to explore weighted aggregation schemes that account for concept frequency and reliability. We expect these extensions to further strengthen the methodological soundness of SPKGDIAG.
>
> >**W3: Several baselines are missing results (“–”) without explanation, leaving unclear whether they failed to run, were incompatible, or underperformed. Furthermore, the framework’s dependence on proprietary GPT-4.1 and text-embedding-3-large raises reproducibility concerns. No analysis examines how alternative LLMs or embedding models affect performance. Given this reliance, a sensitivity or stability study (model substitution, variance, confidence intervals) is necessary to support the robustness claim.**
>
> We appreciate the reviewer’s comments on missing baselines and reproducibility.
>
> Firstly, we added in Table 1: “Entries marked with “–” indicate cases where neither comparable reported results nor runnable official code are available under our experimental setting.”
>
> Secondly, we selected GPT-4.1 due to its advanced contextual understanding of medical dialogues, which we found critical for accurate implicit symptom extraction. While we did not include results for other LLMs due to computational constraints, we acknowledge the value of this analysis and plan to explore alternative LLMs, including domain-specific ones, in future iterations.

---

> ### Author Response · Authors · 2025-11-21
>
> >**W4: Edges are defined solely by symptom overlap. If the global graph is built before data splitting, test nodes could access training information through shared neighbors. The authors should clarify whether graphs are reconstructed within each split and whether any safeguards (e.g., split-specific adjacency or edge regularization) are applied to prevent leakage.**
>
> We thank the reviewer for raising this important point regarding potential data leakage via shared symptom-based edges. We explicitly clarify this graph construction protocol in the revised version of the paper (in **Appendix A.1 Datasets**). We first split the dataset into training and testing subsets, and then built separate patient graphs for each split. Specifically, the training graph is constructed using only training samples, and all model parameters are learned solely on this training graph. During inference, we construct a test graph independently, using only the test set patients and their symptom embeddings. There is no overlap in nodes or edges between the training and test graphs, and no shared neighbors are allowed across splits. This ensures a strict separation and prevents any information leakage.
>
> ---
> We sincerely appreciate the reviewer’s constructive feedback and insightful comments. We hope that the revisions adequately address the concerns raised, and we remain happy to provide any further clarification or additional details if needed.

---

### Official Review · Reviewer_cvYy · 2025-11-01

**Soundness:** 3
**Presentation:** 3
**Contribution:** 2
**Rating:** 4
**Confidence:** 4

**Summary:**

This paper proposes SPKGDIAG, a framework that combines Large Language Models (LLMs) and Graph Neural Networks (GNNs) for automated medical diagnosis. It uses an LLM (GPT-4.1) to extract explicit and implicit symptoms from patient and doctor dialogues, builds a patient-centric knowledge graph based on shared symptoms, and applies a Message Passing Neural Network (MPNN) with two-hop sampling for diagnosis prediction. By integrating language understanding with structured reasoning, SPKGDIAG achieves state-of-the-art results on multiple benchmark and real-world datasets.

**Strengths:**

1. The paper effectively combines LLM-based symptom extraction with graph-based reasoning, bridging unstructured language understanding and structured medical knowledge.
2. Constructing a patient-level knowledge graph captures inter-patient symptom similarities, enhancing interpretability and clinical relevance.
3. Extensive experiments on both public and real-world datasets demonstrate consistent and significant improvements over strong baselines.

**Weaknesses:**

1. The framework exclusively uses GPT-4.1, with no experiments evaluating the impact of different LLMs or model variants on performance.
2. The symptom extraction process fully relies on GPT-4.1 without a clear evaluation of extraction accuracy or hallucination errors, which may introduce noise into the knowledge graph.
3. The paper does not provide any analysis of the time or computational cost associated with constructing and maintaining the patient-centric knowledge graph; please clarify the overall time efficiency of this process.
4. Although the abstract claims interpretability, the main text provides only indirect visualization results without a clear case study or explanation of how the model’s decisions can be interpreted.

**Questions:**

See the above **Weaknesses**.

---

> ### Author Response · Authors · 2025-11-21
>
> >**W1: The framework exclusively uses GPT-4.1, with no experiments evaluating the impact of different LLMs or model variants on performance.**
>
> We appreciate the suggestion. We selected GPT-4.1 due to its advanced contextual understanding of medical dialogues, which we found critical for accurate implicit symptom extraction. While we did not include results for other LLMs due to computational constraints, we acknowledge the value of this analysis and plan to explore alternative LLMs, including domain-specific ones, in future iterations.
>
> >**W2: The symptom extraction process fully relies on GPT-4.1 without a clear evaluation of extraction accuracy or hallucination errors, which may introduce noise into the knowledge graph.**
>
> We are sincerely grateful to the reviewer for this insightful and crucial point. The symptom extraction process is guided by a carefully designed medical prompt (**Appendix A.6**), which reduces hallucinations by constraining outputs to clinically valid symptom-disease pairs. In future work, we aim to benchmark extraction quality against expert-annotated datasets or use metrics such as precision/recall to evaluate hallucination rates.
>
> >**W3: The paper does not provide any analysis of the time or computational cost associated with constructing and maintaining the patient-centric knowledge graph; please clarify the overall time efficiency of this process.**
>
> This is a deeply insightful and important question. As noted in **Section 4.1** and **Appendix A.3**, SPKGDIAG is designed for time and memory efficiency. The patient-centric knowledge graph is sparse by construction (based on symptom overlap), and we use a similarity-based 2-hop subgraph sampling strategy to enable localized, computationally efficient graph construction. This avoids the need to process the full graph and supports scalable mini-batch training using PyTorch Geometric with optimized sparse tensor operations. All experiments were conducted on a workstation equipped with an NVIDIA RTX A5000 GPU and a 16-core CPU, demonstrating the framework's suitability for large-scale, real-world applications.
>
> >**W4: Although the abstract claims interpretability, the main text provides only indirect visualization results without a clear case study or explanation of how the model’s decisions can be interpreted.**
>
> Thank you for highlighting this important point. We clarify that interpretability in SPKGDIAG is demonstrated through multiple complementary analyses. In **Appendix A.4**, we present a detailed case study (Node 79, MZ-4 dataset), where we trace how the model integrates information from neighbors with varying degrees of clinical relevance. We visualize message propagation (**Figure 6**), quantify influence via message norms (**Table 7, Figure 7**), and explain decision rationale through shared symptom profiles. Additionally, we include t-SNE plots (**Appendix A.5, Figure 9**) to show how the model separates patient representations in the latent space, revealing clear class-specific clustering. Confusion matrices (**Figure 10**) further highlight the model’s per-class decision patterns and potential sources of misclassification. Together, these visualizations provide both instance-level and global insight into how SPKGDIAG makes predictions, supporting its interpretability claims beyond accuracy alone.
>
> ---
> We would like to extend our sincere appreciation to the reviewer for their thoughtful and detailed feedback. Your comments have important improvements that enhance the overall contribution of the work. We trust that our revised analyses and explanations adequately address your questions, and we stand ready to provide any further clarification if needed.

---

### Official Review · Reviewer_2Rkj · 2025-11-01

**Soundness:** 2
**Presentation:** 3
**Contribution:** 2
**Rating:** 6
**Confidence:** 2

**Summary:**

This paper presents SPKGDIAG, a framework for automated diagnosis that integrates large language models (LLMs) with patient-centric knowledge graphs (KGs) to support symptom-driven disease prediction through multi-hop similarity message passing. The work explores the intersection of natural language understanding and structured graph reasoning, aiming to improve interpretability and clinical relevance in diagnostic systems.
The framework consists of several key modules:
(1) a symptom extractor that uses an LLM to identify both explicit and implicit symptoms from patient-doctor dialogues;
(2) semantic embedding fusion, which encodes symptom text into high-dimensional embeddings and aggregates them into patient representations;
(3) patient-centric KG construction, where patients are represented as nodes connected by shared symptoms, with 2-hop neighborhood sampling used to expand local subgraphs for downstream learning.

**Strengths:**

1.The paper is well written and generally easy to follow, with a logical structure, supportive visualizations (e.g., Figure 2), and clear mathematical descriptions.

2. SPKGDIAG reports best empirical results on four benchmark datasets (MZ-4, MZ-10, Dxy, and Synthetic), suggesting that the proposed approach is competitive with existing methods.

3.The visualization experiments are informative and help illustrate how the model performs across different datasets.

**Weaknesses:**

1.While Section 4.3 reports ablations on sampling and hop depth, it omits a focused evaluation of the KG’s role. Without a version of the model that excludes the KG (e.g., relying solely on LLM embeddings), it is difficult to assess whether improvements are genuinely due to graph reasoning or merely reflect LLM-derived representations. In addition, since the KG is built from LLM-extracted symptoms, potential hallucinations or extraction noise could distort its structure, an issue that warrants empirical investigation.

2.The edge definition in the patient-centric KG, connecting nodes if they share even a single symptom (Equation 3), risks generating excessively dense graphs, especially given the prevalence of common symptoms like fever or cough in medical datasets. The paper lacks statistics on average patient degree or graph sparsity (e.g., density metrics), leaving open the question of whether the GNN truly leverages local structure or degenerates to global averaging.

3.Table 3 in Section 4.3.1 shows that the MPNN-based SPKGDIAG outperforms the GAT-based variant (SPKGDIAG-GAT) by 10.13% without 2-hop subgraph expansion. This counterintuitive result is puzzling, as GAT theoretically offers greater expressiveness via learnable neighbor attention weights. The authors provide no experimental or theoretical analysis (e.g., overfitting on sparse subgraphs, noisy attention due to loose edges, or MPNN's self-projection advantages) to explain this gap, which weakens claims about the MPNN design's superiority.

**Questions:**

See Weaknesses part.

---

> ### Author Response · Authors · 2025-11-21
>
> >**W1: While Section 4.3 reports ablations on sampling and hop depth, it omits a focused evaluation of the KG’s role. Without a version of the model that excludes the KG (e.g., relying solely on LLM embeddings), it is difficult to assess whether improvements are genuinely due to graph reasoning or merely reflect LLM-derived representations. In addition, since the KG is built from LLM-extracted symptoms, potential hallucinations or extraction noise could distort its structure, an issue that warrants empirical investigation.**
>
> We thank the reviewer for raising the need for a more explicit analysis of the KG’s contribution and the robustness of LLM-based symptom extraction.
>
> - **Role of the patient-centric KG vs. LLM embeddings**: Section 4.3 currently focuses on ablations related to sampling strategy and hop depth. We agree that a clearer evaluation of the KG’s role would strengthen the paper. Although several baselines in **Table 1** do not use a patient graph and therefore already represent “non-KG” models, they differ from our approach in multiple architectural aspects.
>
> - **Influence of LLM extraction noise and hallucinations**: We also agree that symptom extraction errors from the LLM may affect the KG structure. In our design, the KG is built from sets of extracted symptoms: edges are added only when two patients share at least one symptom and are among each other’s top-k most similar neighbors. This helps reduce the impact of rare hallucinations, which are unlikely to be shared across many patients and thus tend not to create many edges.
>
> >**W2: The edge definition in the patient-centric KG, connecting nodes if they share even a single symptom (Equation 3), risks generating excessively dense graphs, especially given the prevalence of common symptoms like fever or cough in medical datasets. The paper lacks statistics on average patient degree or graph sparsity (e.g., density metrics), leaving open the question of whether the GNN truly leverages local structure or degenerates to global averaging.**
>
> We appreciate the reviewer’s insightful comment regarding the potential risk of generating excessively dense graphs due to the edge definition in our patient-centric knowledge graph (KG), which connects nodes based on shared symptoms. To address this concern, we have included detailed structural statistics for all datasets used in our experiments in **Table 4 (Appendix A.1)**, including the average patient degree and graph density—two widely accepted metrics for characterizing graph sparsity. These metrics allow us to quantitatively assess whether the resulting graphs are overly dense or maintain clinically realistic sparsity. Importantly, we observe that datasets such as VNPT, which contain a large and diverse set of patients and symptoms, exhibit low density values (0.034–0.036), confirming that the graphs remain sparse and are far from fully connected. Even in smaller curated datasets like MZ-4 and Dxy, while the densities are comparatively higher (up to 0.333), they still reflect reasonable connectivity levels given the limited and more specific symptom sets.
>
> Furthermore, the average degree varies significantly across datasets, from ~21 in MZ-10 (test) to ~692 in VNPT (train), providing further evidence that the graphs are not uniformly dense. These statistics validate that our edge definition does not universally result in graph over-connectivity. Instead, they demonstrate that the GNN models operate in structurally diverse environments, preserving local neighborhood information in sparser graphs while scaling to denser settings when appropriate. Therefore, the GNN does not simply degenerate to global averaging but continues to leverage local graph structure in clinically meaningful ways.

---

> ### Author Response · Authors · 2025-11-21
>
> >**W3: Table 3 in Section 4.3.1 shows that the MPNN-based SPKGDIAG outperforms the GAT-based variant (SPKGDIAG-GAT) by 10.13% without 2-hop subgraph expansion. This counterintuitive result is puzzling, as GAT theoretically offers greater expressiveness via learnable neighbor attention weights. The authors provide no experimental or theoretical analysis (e.g., overfitting on sparse subgraphs, noisy attention due to loose edges, or MPNN's self-projection advantages) to explain this gap, which weakens claims about the MPNN design's superiority.**
>
> We are sincerely grateful to the reviewer for this insightful point. To address this, we conducted an ablation study (**Figure 3**) to evaluate the contributions of two key components in our model: the Self-Projection mechanism and the aggregated message passing (Agg. Message) in the MPNN. This study compares the full SPKGDiag model with two ablated versions, one without Self-Projection and one without the message aggregation module. The results demonstrate that both components are crucial to the model’s performance. Removing the Self-Projection, which preserves rich patient-specific features via residual connections, leads to a noticeable drop in accuracy across all datasets, particularly in MZ-10 and Dxy, where the neighborhood information is sparse or noisy. Similarly, eliminating the aggregated message degrades performance, showing that incorporating neighbor information is also essential. The full model (SPKGDiag) consistently outperforms the ablated variants, confirming that self-information retention and neighborhood aggregation are complementary and jointly contribute to improved diagnostic accuracy. This supports our design choices and underscores the model's robustness under various data conditions.
>
> Although GAT is theoretically more expressive, in our setting the MPNN has two practical advantages:
>
> - **Strong self-information preservation**: Our node features are already rich patient embeddings from the LLM. The MPNN explicitly applies a self-projection with a residual connection, which preserves this informative representation and adds a controlled neighbor contribution. In contrast, GAT jointly reweights self and neighbors; with small or noisy neighborhoods, this can unintentionally downweight the useful self-feature and amplify weak neighbors.
>
> - **Robustness to sparse and noisy neighborhoods**: Without 2-hop similarity-based expansion, many 1-hop neighborhoods are small and partially induced by very common symptoms. In this mechanism, attention weights in GAT are estimated from a limited and noisy context, which can lead to unstable or overfitted behavior. Degree-normalized aggregation in the MPNN behaves more like a smooth, regularized operator and is empirically more robust. When 2-hop similarity-based sampling is enabled, GAT performance improves and the gap to MPNN narrows, which supports this explanation.
>
> ---
> We express our sincere thanks to the reviewer for their rigorous and thoughtful review. Your feedback has led to notable improvements in the precision and completeness of the paper. We hope that our responses and updated results satisfactorily resolve the issues identified, and we would welcome any additional queries.

---

### Official Review · Reviewer_5wF3 · 2025-11-03

**Soundness:** 2
**Presentation:** 2
**Contribution:** 2
**Rating:** 4
**Confidence:** 4

**Summary:**

This paper proposes SPKGDIAG, a framework for automated medical diagnosis. The authors argue that existing methods fail to adequately model patient-to-patient similarities and struggle to integrate structured knowledge with the capabilities of Large Language Models (LLMs). The proposed framework utilizes LLMs to extract information from the patient conversation and convert the information into text embeddings. Then a patient-centric knowledge graph is constructed, so that a MPNN module can be used to perform node classification to predict the disease of the target patient.

**Strengths:**

1. The high-level idea of reframing diagnosis as a node classification problem on a patient-patient similarity graph is conceptually novel.
2. The overall presentation and the writing is easy to follow and clear.
3. The paper evaluates its method against a very comprehensive and challenging set of modern baselines, including RL, Transformer, and other graph-based approaches (Table 1).

**Weaknesses:**

1. The paper contains a lot of over-claims. For example, the so-claimed "Symptom extractor" is merely a prompt to LLMs to summarize and extract the information from the dialogues.
2. The patient-centric KG construction is simply built on noisy and naive function, where any two patients who share any single symptom would be connected in the KG, which do not make any sense.
3. The patient embeddings are also simply the average of patient's symptom embeddings into a single vector, which igores all granularity and the severity of different symptoms.
4. The framework requires constructing a graph where every patient is a node. The paper does not address the computational and memory cost of building and processing this graph for a real-world hospital system with millions of patients.

**Questions:**

See weaknesses above.

---

> ### Author Response · Authors · 2025-11-21
>
> >**W1: The paper contains a lot of over-claims. For example, the so-claimed "Symptom extractor" is merely a prompt to LLMs to summarize and extract the information from the dialogues.**
>
> Thank you for the valuable feedback. We acknowledge that the symptom extractor module leverages prompting strategies with GPT-4.1 for extracting symptom-related information from dialogues. However, we respectfully clarify that the term "Symptom Extractor" is used not to suggest a novel model architecture, but rather to denote a critical component within the overall SPKGDIAG framework. Our contribution lies in the integration of this LLM-based extraction with a downstream semantic embedding pipeline, enabling structured symptom representation and further graph-based reasoning for diagnosis. We also designed prompt templates with constraints (**Appendix A.6**) to ensure the clinical relevance and consistency of the extracted symptoms. Hence, the component functions as a principled and effective mechanism, beyond mere summarization.
>
> >**W2: The patient-centric KG construction is simply built on noisy and naive function, where any two patients who share any single symptom would be connected in the KG, which do not make any sense.**
>
> We appreciate this observation and would like to offer further clarification. The adjacency function initially connects patients with at least one shared symptom to ensure inclusiveness in the first layer of the KG. However, this is only the first step of a multi-step refinement. The similarity-based k-hop neighborhood sampling (**Section 3.4, Equations 5–6**) further filters and ranks neighbors based on cosine similarity of semantic embeddings, ensuring that only clinically relevant and semantically similar connections are preserved in the subgraph used for message passing. Ablation studies (**Table 3, Section 4.3.1**) empirically show that this multi-hop similarity sampling substantially improves diagnostic performance across all datasets, confirming the effectiveness of the constructed graph structure.

---

> ### Author Response · Authors · 2025-11-21
>
> >**W3: The patient embeddings are also simply the average of patient's symptom embeddings into a single vector, which igores all granularity and the severity of different symptoms.**
>
> Thank you for raising this point. The averaging strategy is employed as a practical and interpretable method for generating initial patient embeddings. While we use mean pooling to generate patient-level symptom embeddings for computational efficiency and vector normalization, this representation serves only as a first-stage input. We recognize that this method does not explicitly model symptom severity; however, our framework compensates for this limitation in multiple ways:
> - The symptom embeddings are derived from context-aware LLM-based encodings, which capture rich semantic nuances including severity indicators such as intensity adjectives (e.g., “mild” vs. “severe”) that are preserved in the vector space.
> - The downstream MPNN module (**Section 3.5**) enhances this embedding by integrating neighborhood context via message passing and self-projection, effectively reintroducing symptom-level granularity and local contextualization during training.
> - Moreover, we apply multi-hop neighbor sampling to enrich each node’s context beyond individual symptoms, balancing local and global representation.
> That said, incorporating explicit modeling of symptom severity and temporal dynamics remains an important direction for future work (**Section 5 – Conclusion**).
>
> >**W4: The framework requires constructing a graph where every patient is a node. The paper does not address the computational and memory cost of building and processing this graph for a real-world hospital system with millions of patients.**
>
> We thank the reviewer for this critical perspective. While our current work focuses on research-scale and semi-large datasets (e.g., VNPT with 230K patients), we agree that scalability is essential for real-world deployment. We would like to highlight the following:
> - The graph is sparsely constructed, as connections are only made between patients sharing symptoms (**Equation 3**), and further restricted via top-k similarity-based neighborhood sampling, maintaining low memory overhead.
> - We do not perform full-graph message passing; instead, we compute localized 2-hop subgraphs per patient (**Section 3.4**), which allows for parallelized mini-batch training and avoids the need for full adjacency matrix storage.
> - Our implementation leverages PyTorch Geometric, which efficiently supports sparse mini-batch GNN training on large-scale graphs (**Appendix A.3**).
>
> To better support large-scale hospital settings, we are exploring dynamic graph sampling, graph pruning, and index-based subgraph caching in future work, as mentioned in the conclusion.
>
> ---
> Once again, we are grateful for the reviewer’s thorough evaluation and insightful observations. The suggestions offered have guided meaningful refinements to our methodology and empirical analysis. We hope that the clarifications and new results presented herein satisfactorily resolve the concerns raised. We would be pleased to engage in any further discussion.

---

### Meta-Review · Area_Chair_Xppm · 2026-01-07

**Summary:**

1. Reviewers question if connecting patients based on shared symptoms will lead to noisy, dense graphs. Plus, simply averaging symptom embeddings into a patient vector ignores critical details.
2. Many concerns about the experiments, such as the lack of a "no-graph" ablation study to prove the graph structure is actually necessary. Also, the evaluation metrics were considered limited.
3. Some concerns about the task formulation as a classification problem.
4. Some concerns about the reliance on a single proprietary LLM.

**Reviewer Concerns:**

The authors add a "no-graph" ablation study to show that the performance will drop without the graph structure. This mitigate the second concern a little bit. However, the other concerns about graph construction method, task formulation, reliance on single LLM still hold,

**Reviewer Scores:**

Reviewer 5wF3: likely not change the score.

Reviewer 2Rkj: likely not change the score.

Reviewer cvYy: likely not change the score.

Reviewer cfcg: likely not change the score.

Reviewer gLUz: not change the score as the reviewer explicitly says so

Reviewer qS3q: maybe increase by 1 or 2 because the reviewer says "I'll take your feedback into consideration".

---

### Decision · Program_Chairs · 2026-01-26

Reject